# Estimation of NO₂ emission strengths over Riyadh and Madrid from space from a combination of wind-assigned anomalies and machine learning technique

Qiansi Tu[1,2], Frank Hase[2], Zihan Chen[3], Matthias Schneider[2], Omaira García[4], Farahnaz Khosrawi[2], Shuo Chen[5], Thomas Blumenstock[2], Fang Liu[6], Kai Qin[7], Jason Cohen[7], Qin He[7], Song Lin[1], Hongyan Jiang[1,8], Dianjun Fang[1,9]

[1]Tongji University, School of Mechanical Engineering, Shanghai, China
[2]Karlsruhe Institute of Technology (KIT), Institute of Meteorology and Climate Research (IMK-ASF), Karlsruhe, Germany
[3]Karlsruhe Institute of Technology (KIT), Department of Informatics, Karlsruhe, Germany
[4]Izaña Atmospheric Research Centre (IARC), Meteorological State Agency of Spain (AEMet), Tenerife, Spain
[5]Z-one tech Co., Ltd., Shanghai, China
[6]Beijing Chehejia Automobile Technology Co., Ltd., Beijing, China
[7]China University of Mining and Technology, School of Environment and Spatial Informatics, Jiangsu, China
[8]Jiangsu University of Science and Technology, School of Mechatronic and Power Engineering, Zhenjiang, China
[9]Qingdao Sino-German Institute of Intelligent Technologies, Qingdao, China

*Correspondence to*: Matthias Schneider (matthias.schneider@kit.edu), Dianjun Fang (fang@tongji.edu.cn)

**Abstract.** Nitrogen dioxide (NO₂) air pollution provides valuable information for quantifying NOx (NOx = NO + NO₂) emissions and exposures. This study presents a comprehensive method to estimate average tropospheric NO₂ emission strengths derived from four-year (April 2018 – June 2022) TROPOMI observations by combining a wind-assigned anomaly approach and a Machine Learning (ML) method, the so-called Gradient Descent. This combined approach is firstly applied to the Saudi Arabian capital city Riyadh, as a test site, and yields a total emission rate of $1.04 \times 10^{26}$ molec./s. The ML-trained anomalies fit very well with the wind-assigned anomalies with an $R^2$ value of 1.0 and a slope of 0.99. Hotspots of NO₂ emissions are apparent at several sites where the cement plant and power plants are located and over areas along the highways. Using the same approach, an emission rate of $1.80 \times 10^{25}$ molec./s is estimated in the Madrid metropolitan area, Spain. Both the estimate and spatial pattern are comparable to the CAMS inventory.

Weekly variations of NO₂ emission are highly related to anthropogenic activities, such as the transport sector. The NO₂ emissions were reduced by 24% at weekends in Riyadh, and high reductions are found near the city center and the areas along the highway. An average weekend reduction estimate of 30% in Madrid is found. The regions with dominant sources are located in the east of Madrid, where the residential areas and the Madrid-Barajas airport are located. Additionally, the NO₂ emissions decreased by 21% in March-June 2020 compared to the same period in 2019 induced by the COVID-19 lockdowns in Riyadh. A much higher reduction (60%) is estimated for Madrid where a very strict lockdown policy was implemented. The high emission strengths during lockdown only persist in the residential areas and cover smaller areas during weekdays than at

weekends. The spatial patterns of NO$_2$ emission strengths during lockdown are similar to those observed at weekends in both

cities. Though our analysis is limited to two cities as testing examples, the method has proved to provide reliable and consistent results. It is expected to be suitable for other trace gases and other target regions. However, it might become challenging in some areas with complicated emission sources and topography, and specific NO$_2$ decay times in different regions and seasons should be taken into account. These impacting factors should be considered in the future model to further reduce the uncertainty budget.

**1 Introduction**

Nitrogen oxides (NOx = NO + NO$_2$) are a group of highly reactive trace gases (NO and NO$_2$). NOx are toxic to human health and play a key role in tropospheric chemistry by catalyzing tropospheric O$_3$ formation and acting as aerosol precursors, and this tropospheric O$_3$ is a secondary pollutant that is also harmful to human health (IPCC, 2021). The emission of NOx is dominated by human activities and is mostly related to fossil fuel or biomass combustion (Goldberg et al., 2019). The major

anthropogenic source in Europe is road transport (39%), followed by another four sectors with similar shares: energy production and distribution (14%), commercial, institutional, and households (13%), energy use in industry (11%) and agriculture (11%) (EEA, 2021). The near-surface abundance of NOx has generally increased with urbanization and industrialization (IPCC, 2021; Barré et al., 2021). Additionally, due to its short tropospheric lifetime (1 – 12 h) (Beirle et al., 2011; Stavrakou et al., 2013), NOx concentrations are highly variable and strongly correlated with local emission sources

(Goldberg et al., 2019). Thus, NO$_2$ observations can be considered as an excellent indicator to NOx emissions. The accurate knowledge of spatial and temporal distribution of NO$_2$ atmospheric abundances, for this reason, is critical.

Space missions succeed in delivering well-resolved maps of tropospheric NO$_2$ columns, from the early Global Ozone Monitoring Experiment (GOME) (Burrows et al., 1999), to the widely used Ozone Monitoring Instrument (OMI) (Boersma et al., 2007; He et al., 2021), to the latest TROPOspheric Monitoring Instrument (TROPOMI) (Veefkind et al., 2012). Among

them, TROPOMI onboard Sentinel-5 Precursor (S-5P) since October 2017 has an outstanding importance. It is a push broom grating spectrometer, and measures direct and reflected sunlight in ultraviolet, visible, near-infrared, and shortwave infrared bands (Veefkind et al., 2012). TROPOMI offers daily coverage of data with an unprecedented spatial resolution of $3.5 \times 7$ km$^2$ ($3.5 \times 5.5$ km$^2$ since August 2019) and a high signal-to-noise ratio (Copernicus Sentinel-5P, 2018; van Geffen et al., 2021). The TROPOMI NO$_2$ data have been used for a variety of studies to estimate the NOx lifetime and emissions. For example,

Lorente et al. (2019) has demonstrated that the strength and distribution of NO$_2$ emissions from Paris can be directly determined from the TROPOMI NO$_2$ measurements. Beirle et al. (2019) mapped the NOx emissions on high spatial resolution based on the continuity equation and quantified urban pollution from Riyadh, Saudi Arabia (8.5 kg/s over $250 \times 250$ km$^2$). A top-down NOx emission estimate approach was developed by Goldberg et al. (2019) and it reported that three megacities (New York City, Chicago, and Toronto) in North America emitted 3.9 – 5.3 kg/s NOx. Liu et al. (2020) demonstrated a 48% drop in the

tropospheric NO$_2$ column densities in China during the COVID-19 lockdown. The reductions of NO$_2$ emission across the

European urban areas resulting from the lockdown were studied by Barré et al. (2021) and -23% changes on average were obtained based on TROPOMI $NO_2$ observations.

TROPOMI is unique due to its very high spatial- and temporal- resolution, which provides a large amount of data despite a planned mission lifetime of only about four years. This huge data set offers a possibility for its exploitation by the quickly
developed artificial intelligence – Machine Learning (ML) techniques. For example, the application of ML to assess the $NO_2$ pollution changes during the COVID19 lockdown (Petetin et al., 2020; Keller et al., 2021; Barré et al., 2021; Chan et al., 2021). However, most studies focus on changes in $NO_2$ column abundances. The accurate amount and spatial pattern of deduced emission strengths are also important and can help air quality policy development.

In this study, the Gradient Descent (GD) approach in ML incorporating the wind-assigned method (Tu et al., 2022a, 2022b) is
used to train the "modeled truth" constructed from a simple downwind plume model for the emissions on each grid pixel using space borne $NO_2$ observations, to estimate the $NO_2$ emission strengths of two (mega)cities: Riyadh (Saudi Arabia) and Madrid (Spain). The paper is organized as follows. Sect. 2 presents the data set and the combined method (wind-assigned and ML methods). The approach will be first applied to the Saudi Arabian capital city Riyadh for its evaluation and then applied to Madrid, followed by the discussion of the differences on weekdays and at weekends, and the changes before and during the
COVID-19 lockdown period (Sect. 3). Conclusions are given in Sect. 4.

## 2 Data and Methodology

### 2.1 TROPOMI tropospheric $NO_2$ columns and wind data

The $NO_2$ data used in this study are obtained from the Sentinel-5P Pre-Operations Data Hub (https://s5phub.copernicus.eu/dhus/#/home), which provides level 2 datasets with three different data streams: the Non-Time
Critical or Offline (OFFL), the Reprocessing (RPRO) and the near-real-time (NRTI) streams. The NRTI is available within 3 h after the actual satellite measurement and may sometimes be incomplete and has a slightly lower data quality (http://www.tropomi.eu/data-products/level-2-products, last access: 14 September 2022), and thus, this data set is not considered here. The RPRO data covers a time range from 30 April 2018 – 17 October 2018, and the OFFL data covers the remaining time period. Meanwhile, the $NO_2$ dataset is an aggregate of different versions. The RPRO data is v1.2, while OFFL
includes several versions: v1.2 until March 20, 2019, v1.3 until 29 November 2020, v1.4 until 5 July 2021, v2.2 until 15 November 2021, and v2.3 until 17 July 2022 and v2.4 afterwards. An improved FRESCO cloud retrieval has been introduced in v1.4, which leads to higher tropospheric $NO_2$ columns over areas with pollution sources under small cloud coverage (van Geffen et al., 2022). We use the TROPOMI tropospheric $NO_2$ from May 2018 to June 2022 over Saudi Arabian capital city of Riyadh and another (mega)city in Europe, such as Madrid, Spain. The quality flag (qa_value) is recommended to be >0.75,
with which data are restricted to cloud-free (cloud radiance fraction < 0.5), and snow-ice-free observations (van Geffen et al., 2021). There are nearly 1,380,000 in Riyadh (23.6ºN – 25.4ºN; 46.1ºE – 47.4ºE) and 930,000 measurements in Madrid (39.5ºN

– 41.5ºN; 4.5ºW – 3ºW) of good quality over three years. These observations are then binned for this study on a regular 0.1° × 0.1° grid, using as prerequisite that the number of observations is larger than 5 at the respective grid point. The amounts of TROPOMI measurements in each 0.1° grid pixel is distributed evenly with a number range of 4400-4800 in Riyadh, whereas larger differences are observed in Madrid with a number range of 2200-3700 (see Figure A- 1).

We use the horizontal wind information from the ERA5, which is the fifth-generation climate reanalysis produced by the European Centre for Medium-Range Weather Forecasts (ECMWF) at a spatial resolution of 0.25° × 0.25° (Copernicus Climate Change Service, 2017). $NO_2$ is a short-lived species, following the orography. Therefore, we use ERA5 at 10 m (Figure A- 2).

## 2.2 Wind-assigned and ML methods

The averaged distribution of emitted $NO_2$ over a long-term period can be approximated by an evenly distributed cone-shape plume, which is prescribed by wind speed and direction, and source strength with consideration of its temporal decay:

$$\Delta NO_{2\ (xi,yi)} = \frac{\varepsilon}{v \cdot d_{(xi,yi)} \cdot \alpha} \times exp\left(-\frac{t}{\tau}\right)$$
Eq. (1)

where $\varepsilon$ is the emission strength and has an initialized value of $1\times10^{26}$ molec./s. The study area is binned on a regular 0.1° × 0.1° grid and the emission rates at each grid are assumed to be constant during the study period. $\alpha$ is the angle of the emission cone and has an empirical value of 1/3 rad (i.e., 60°) (Tu et al., 2022a). $d$ and $t$ are the distance in m and transport time in hour between the downwind location and $NO_2$ emission source, respectively. $v$ is the wind speed in m/s from ERA5 and $\tau$ is the lifetime/decay time in hour for $NO_2$. For simplification, seasonal and spatial variability of lifetime is not considered, and empirical values based on Beirle et al. (2019, 2011), i.e., fixed values of 4 hours for Riyadh and 7 hours for Madrid, are used in this study. The daily plumes ($\Delta NO_2$) from the individual emission source are computed based on Eq. (1) and then are super-positioned to have a total daily plume. The ERA5 model wind is divided into two opposite wind regimes based on the predominant wind regimes in each site (i.e., S: 90°-270° and N: the rest for Riyadh; SW: 135°-315° and NE: the rest for Madrid, see Figure A- 2). A temporally averaged $\Delta NO_2$ plume is obtained for each wind regime and the difference between the two plumes generates the wind-assigned anomalies (for more details see Tu et al., 2022a, 2022b).

The study area has x × y (=N) grids. Each grid cell is considered as an independent point source at position $(s_{lat_i}, s_{lat_j})$, which yields a map of wind-assigned anomalies ($c_{s_{lat_i}, s_{lat_j}}$). The wind information is assumed to be constant at each time over the study area in this study. The modeled wind-assigned anomalies derived from the point source located at the center grid $(lat_{i_0}, lon_{j_0})$ is considered as a parent map (see Figure A- 3a):

$$c_{s_{lat_{i_0}}, s_{lat_{j_0}}} = (p_{lat_1, lon_1} \cdots p_{lat_i, lon_j} \cdots p_{lat_x, lon_y})$$
Eq. (2)

The anomalies derived from other point source is identical to the parent anomalies, and value at each grid depends on the relative location to the parent one (see Figure A- 3b):

$$c_{s_{lat_i},s_{lon_j}} = (p_{lat_1-lat_{i_0},lon_1-lon_{i_0}} \cdots p_{lat_i-lat_{i_0},lon_j-lon_{i_0}} \cdots p_{lat_x-lat_{i_0},lon_y-lon_{i_0}}) \qquad \text{Eq. (3)}$$

These maps of wind-assigned anomalies at each grid are the inputs for the further step, which needs to be reformatted. The locations of the grids are reordered in the sequence of latitude and longitude values from west to east and from north to south. The first grid at $(lat_1, lon_1)$ locates in the far northwest and the last grid $(lat_x, lon_y)$ locates in the far southeast. Therefore, each map of wind-assigned anomalies is converted to a new column vector $c_k = (a_{k,1} \quad \cdots \quad a_{k,N})^T$, i.e., $a_{k,k}$ represents the wind-assigned anomalies at $k^{th}$ grid cell derived from point sources at $k^{th}$ grid cell. The N grids generate N vectors to construct an N × N matrix:

$$\mathbf{M} = (c_1 \cdots c_N) = \begin{pmatrix} a_{1,1} & \cdots & a_{N,1} \\ \vdots & \ddots & \vdots \\ a_{1,N} & \cdots & a_{N,N} \end{pmatrix} \qquad \text{Eq. (4)}$$

The estimated emission rate is a column vector $w = (w_1 \quad \cdots \quad w_N)^T$. Since the emission rates cannot be negative, we use $\log(w_k)$ as a proxy of the $w_k$. The final result is then the exponent of the $\log(w_k)$ and scaled by the initial $\varepsilon$ of $1 \times 10^{26}$ molec./s. Then the modeled-calculated map ($m$) of the wind-assigned anomalies can be written as:

$$m = \mathbf{M} \times w = \begin{pmatrix} a_{1,1} & \cdots & a_{N,1} \\ \vdots & \ddots & \vdots \\ a_{1,N} & \cdots & a_{N,N} \end{pmatrix} \times (w_1 \quad \cdots \quad w_N)^T = (m_1 \quad \cdots \quad m_N)^T \qquad \text{Eq. (5)}$$

The wind-assigned anomaly method is also applied to the TROPOMI tropospheric $NO_2$ column, yielding to a true map $y = (y_1 \quad \cdots \quad y_N)^T$.

To estimate the emission strengths accurately, the modeled map ($m$) should approximate the true map ($y$). This problem is then converted to find the best $w$ which results in the minimum value of the difference between $y$ and $m$, i.e., the cost function:

$$L(y,m) = \frac{1}{N}\sum_{i=1}^{N}(y_i - m_i)^2 = \frac{1}{N}\sum_{i=1}^{N}(y_i - (a_{1,i} \quad \cdots \quad a_{N,i}) \times (w_1 \quad \cdots \quad w_N)^T)^2 \qquad \text{Eq. (6)}$$

In our approach, the above equation can be considered as solving a linear system with constraints over the coefficients. In the ML framework, the popular GD algorithm can be a simple yet effective solution to find the coefficients. These coefficients can satisfy the approximation and the constraints at the same time, by formulating some of the constraints into the loss function that needs to be optimized. The main idea of GD is to find the partial derivatives of all coefficients in the system with respect to the loss function and use the local (gradient) information to reach the solution closer to the true state, which minimizes the approximation loss. In practice, this is implemented in an iterative process in which the data are sampled for the required gradients. However, there is only one single "data point" (one column vector) in our problem formulation. For each iteration (Eq. (7)), the new weight ($w_{t+1}$) is equal to the old weight ($w_t$) minus the gradient times the learning rate $\eta$ (or so-called step size). Here, we use the default settings ($\eta = 0.001$) as used by Kingma and Ba (2015):

$$w_{t+1} = w_t + \Delta w_t = w_t - \eta \times \sum_{i=1}^{N}\frac{\delta L}{\delta w_t} \qquad \text{Eq. (7)}$$

The selected areas in this study are highly isolated from the neighboring sources and thus, the emission rates at the edge can be assumed to be zero. However, the initialized constrain of them can increase the final biases. Therefore, we use a larger study area with $(n+2) \times (m+2)$ grids as the input data and remove the outmost rectangle within 2-grid width to the target area of $n \times m$ grids.

When applying GD for complicated systems with many parameters, there are many variations of GD that rely not only on the gradients but also introduce additional temporal information, i.e., the accumulation of gradients over time known as "momentum" to help GD converge faster and more reliable. Among those algorithms we decided to use Adaptive Moment Estimation (ADAM), because it is characterized by efficiency, little cost requirement (Kingma and Ba, 2015) compared to second-order methods such as BFGS (Broyden–Fletcher–Goldfarb–Shanno), and for our problem, it can slightly outperform

other GD variations, such as the original gradient descent (GD) with momentum or Adadelta/Adagrad. In addition, it has been documented that it is superior by employing the cumulative first-order and second-order moments and, thus, become the de-facto method in the current deep learning scene when dealing with large data and parameters (Kingma and Ba, 2015).

## 3 Results and Discussion

### 3.1 Approach test for NO$_2$ emissions in Riyadh

Riyadh was chosen as the test site because this city with arid climate, has high NOx emissions due to the high population density (~4,300 residents/km$^2$; https://worldpopulationreview.com/world-cities/riyadh-population, last access: 29 March 2022) and it has punctual strong NOx sources close to the metropolitan area, such as cement plant and power plants. Moreover, Riyadh is remote from other sources, and has favorite weather conditions with low cloud cover and high surface albedo (Beirle et al., 2019; Rey-Pommier et al., 2022). The typical two wind regimes presented in Riyadh favors the applicability of the wind-

anomaly method and is another reason of choosing it for the work.

Figure 1 illustrates the averaged wind-assigned plumes derived from the TROPOMI tropospheric NO$_2$ and ML method over the analyzed period (April 2018 - March 2021). The ML-modelled plumes agree excellently with the satellite's results (true map). A stronger plume is observed in the south of Riyadh, as the wind is more from the north (Figure A- 2). The good correlation between these two maps is also presented in the one-to-one figure with an R$^2$ value of 1.0 and a slope value of 0.99

(Figure 1c). The estimated emission strengths based on the ML model (Figure 1d) show a similar spatial pattern, especially on the main sources near the city center to the results in Beirle et al. (2019) (Fig. 2). Hotspots of NO$_2$ emissions are apparent at several sites where the cement plant and power plants are located and over areas along the highways (Figure 1d). These power plants have capacities larger than 1 GW and use crude oil and partly natural gas as fossil fuels (Beirle et al., 2019). The total emission rate is about $1.04 \times 10^{26}$ molec./s. Our estimate is slightly higher than the Beirle's result ($8.3 \times 10^{25}$ molec./s from

December 2017 to October 2018), in which wind fields from the ECMWF operational analysis at about 450 m above the ground were used (Beirle et al., 2019). The difference might be due to the different study periods and methods used. The

pattern of wind direction is similar at higher level (100 m), while the wind speed increases (Figure A- 2). therefore, it is expected that wind at these levels introduces minor impacts on the estimates.

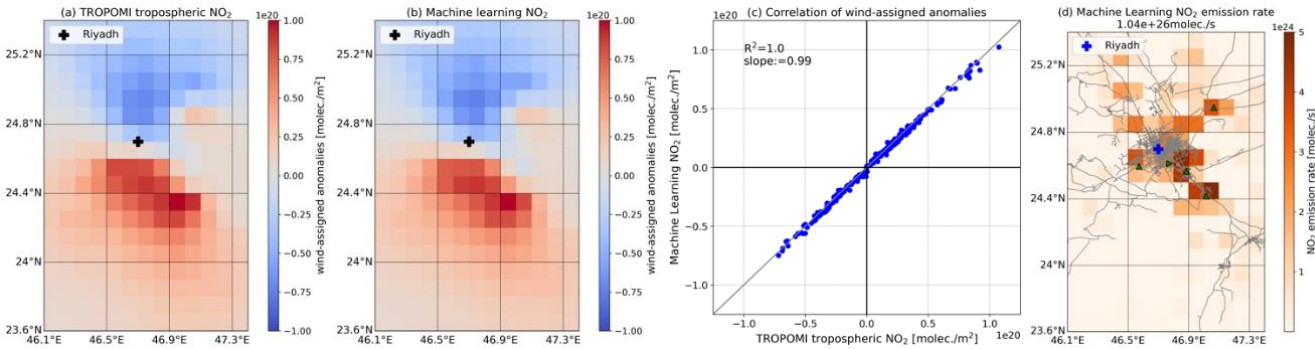

**Figure 1: Wind-assigned plumes derived from (a) TROPOMI tropospheric NO₂ and (b) ML method, (c) correlation plot between (a) and (b) for each grid (the x and y labels represent the data sets from where the wind-assigned anomalies are derived), and (d) estimated emission rates in Riyadh, Saudi Arabia. Data in (a), (b), and (d) are gridded on a regular latitude-longitude grid with 0.1° spacing. In (d) number in the figure's title presents the total emission rate; triangle symbols and right-triangle symbol represent power plants and cement plant, respectively; grey lines represent the highways (data derived from www.openstreetmap.org © 185 OpenStreetMap and www.mapcruzin.com, last access: 11 April 2022).**

### 3.2 NO₂ emission in Madrid

As a (mega)city in Europe, Madrid in Spain is another target in this study. The population of the Madrid metropolitan area is estimated to be about 6.7 million and nearly half of the residents live in Madrid city, resulting in a population density of ~5,400 residents/km$^2$ (https://worldpopulationreview.com/world-cities/madrid-population, last access: 29 March 2022). Figure 2a and 190 b display the wind-assigned anomalies derived from TROPOMI observations and ML method, showing clearly pronounced bipolar plumes which are symmetrical in the Madrid city center. The ML-trained anomalies have a very good agreement with the TROPOMI one with an R$^2$ value of 0.99 and a slope of 0.99 (Figure 2c). The spatial pattern of estimated emission strengths is shown in Figure 2d, which is comparable to that of CAMS-REG-AP (Copernicus Atmospheric Monitoring Service regional anthropogenic emission inventory, https://eccad.aeris-data.fr/catalogue/, last access: 31 March 2022; Granier, et al., 2019; 195 Kuenen, et al., 2021) (Figure 2e). The CAMS-REG-AP covers emissions from the UNECE-Europe for the main air pollutants (e.g., NOx, expressed as NO₂) with a spatial resolution of 0.05° × 0.1° - 0.1° × 0.1° in longitude and latitude on a yearly basis over Europe (Kuenen et al., 2014). The v5.1-BAU2020 is the latest version of a series of emission inventories, which extrapolate CAMS-REG-v5.1 to the year 2020, neglecting the impacts related to Covid-19 (Kuenen et al., 2021). CAMS-REG-v5.1 covers the data from 2000-2018 and v4.2-ry covers the updated recent years 2018 and 2019 200 (https://eccad3.sedoo.fr/#CAMS-REG-AP, last access: 17 August 2022). The total emission rate over the whole study is about 1.80×10$^{25}$ molec./s, close to the CAMS inventory value of 9.2×10$^{24}$ molec./s in 2020.

Our estimate is smaller than a previous estimate of 6.8×10$^{25}$ molec./s derived from the Ozone Monitoring Instrument (OMI) data during 2005-2009 (Beirle et al., 2011). The time series of tropospheric NO₂ observed by OMI since 2004 and TROPOMI since 2018 in two study sites are shown in Figure A- 14 and their correlations are shown in Figure A- 15. NO₂ amounts

increased since 2004 and reached highest value around 2016, except a sudden drop in 2013 in Riyadh. A continuous decrease is observed in Madrid and the COVID lockdown leads to a reduction of $NO_2$ emission in 2020. $NO_2$ concentration retrieved from the OMI observations are generally lower (slope = 0.8074) than TROPOMI results with a mean bias of $6.3\times10^{18} \pm 9.8\times10^{18}$ molec./m$^2$ in Riyadh. The $R^2$ value in Madrid area ($R^2$=0.8542) is slightly smaller than the value in Riyadh ($R^2$=0.9357). However, the mean bias is lower and the standard deviation is higher in Madrid area with a value of $1.9\times10^{18} \pm$

$1.2\times10^{19}$ molec./m$^2$ (slope=0.8353). The ML emission rate retrieved from OMI observations (binned in 0.25°×0.25°) is 17% lower in Riyadh and 18% lower in Madrid area than those from TROPOMI observations. Thus, the discrepancy between this and previous study is mainly due to the data sets used.

Apart from that, it is important highlighting that in the last decades, considerable efforts have been made in promoting the control and regulation of air quality policies across Europe (EEA report, 2020). In this context, Madrid City Council launched

the Air Quality and Climate Change Plan for the city of Madrid (Plan A) in 2017, aiming at reducing pollution and adapting to climate change and ~25% reduction of $NO_2$ concentrations in the central area were assumed by 2020 (https://www.madrid.es/UnidadesDescentralizadas/Sostenibilidad/CalidadAire/Ficheros/PlanAire&CC_Eng.pdf, last access: January 21, 2022). The binned emission rates agree well between the CAMS inventory and the ML-trained results with an $R^2$ value of 0.67 and a slope of 1.16 (Figure 2f). The ML-trained results are higher than the inventory. This is probably related to

the fact that TROPOMI measures real-time $NO_2$ emissions which are not fully considered in the CAMS inventory.

Based on the spatial pattern, the dominant $NO_2$ sources can be easily distinguished. High $NO_2$ emissions are found near the city center, while the highest emissions are occurred to the east, south, and southwest where the residential areas are located. The northwest of Madrid is the natural space and the Guadarrama mountains range runs in the NE-SW direction. Therefore, no obvious $NO_2$ sources can be found in these mountain regions. The Madrid-Barajas airport (presented as the triangle symbol

in Figure 2d and e), which is the main international airport in Spain and the second-largest airport in Europe, is near the northeast of the city center where the region shows high $NO_2$ emissions. This is because aircraft exhaust emissions are highly enriched in $NO_2$ during taxiing and taking off (Herndon et al., 2004) and the near-airport $NO_2$ concentrations are higher than the emissions from highways and busy roadways (Hudda et al., 2020). In addition, the orographic feature, i.e., the development of mountain breezes along the slope of the Guadarrama range causes the accumulation of pollutants in the NE -SW axis (Querol

et al., 2018). Significant plumes of $NO_2$ columns are observed for wind from narrow wind regimes covering $NE_{1/2}$ (0°-90°) and $SW_{1/2}$ (18°-270°) (Figure A- 7(a)-(b)). $NO_2$ accumulates near the city center for $NW_{1/2}$ wind (270°-360°), and a much weaker plume is found for $SE_{1/2}$ (90°-180°) wind regimes due to fewer wind days and weaker wind speed (Figure A- 7(c)-(d)).

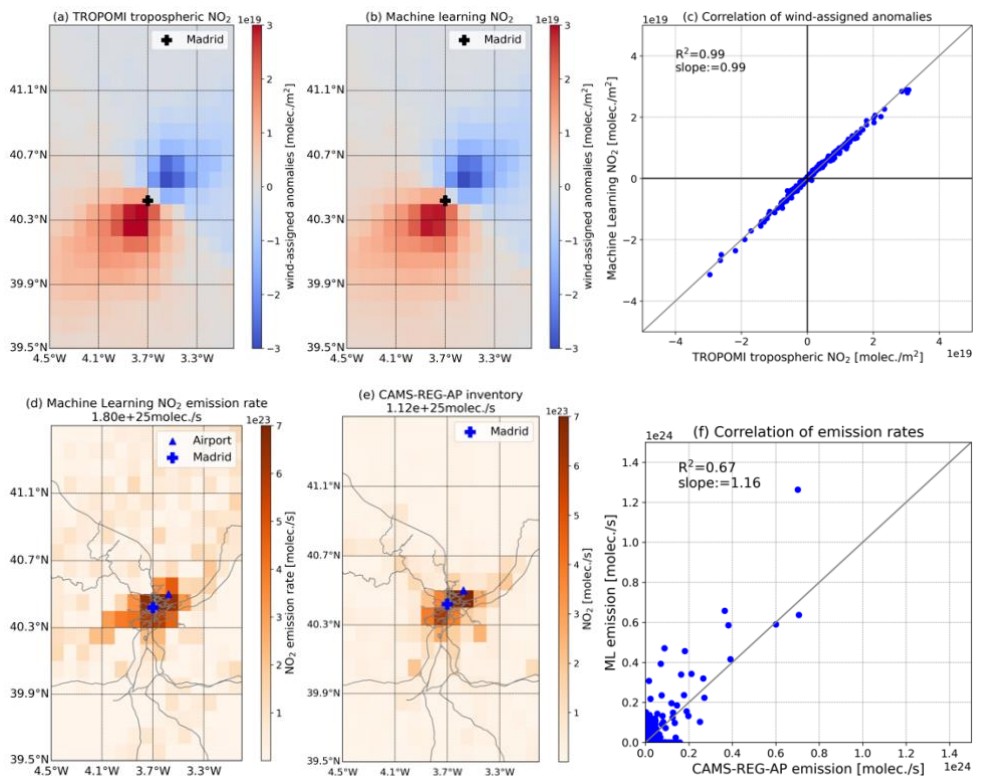

**Figure 2: (a)-(d): same figures as Figure 1, but for the Madrid area, Spain. (e): spatial distribution of CAMS-REG-AP inventory, (f) correlation of emission rates between ML and CAMS-REG-AP inventory.**

### 3.3 NO$_2$ emission changes on weekdays and at weekends

NOx emission variations result in significant changes in the weekly cycle, which is an unequivocal sign of anthropogenic sources (Beirle et al., 2003).

The estimated emission rates for weekdays (Sunday to Thursday) and weekends (Friday and Saturday) in Riyadh are presented in Figure 3. It should be noted that the weekends in Saudi Arabia are Fridays and Saturdays. The lowest NO$_2$ column abundances are observed on Fridays, followed by the ones on Saturdays (Figure A- 8). The NO$_2$ emissions were reduced by 24% at weekends, and high reductions were found near the city center and the areas along Highway 65. Highway 65 is a major north-south highway in central Saudi Arabia and runs in the southeast-northwest direction, connecting Riyadh to Al Majma'ah in the northwest and to Kharj in the southeast (Figure A- 9).

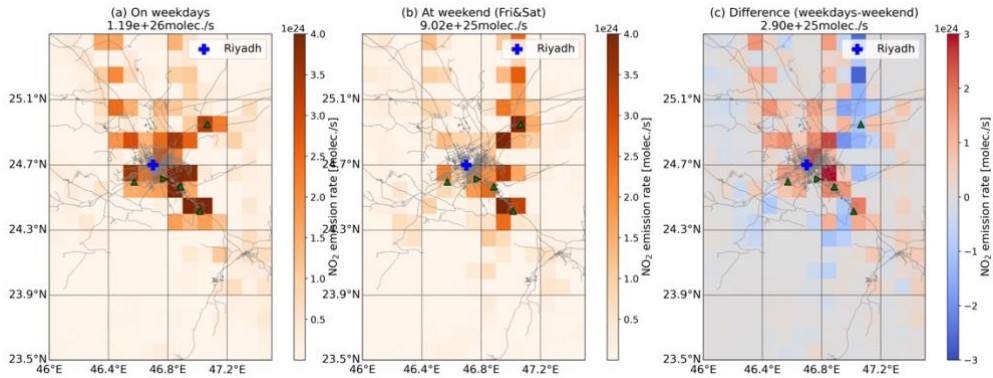

**Figure 3: Averaged ML estimated emission strengths during (a) weekdays (Sunday to Thursday), (b) weekends (Friday and Saturday), and (c) their difference (weekdays-weekend) in the Riyadh. Number in each figure's title presents the total emission rate.**

Significant column declines are found in large cities, especially in Europe, at weekends (Stavrakou et al., 2020). The weekly cycle of NO$_2$ column abundances in the Madrid area is different from that in Riyadh, as the lowest amounts are on Sundays, the second days of weekends (Figure A- 10). An outstanding difference becomes apparent as much higher NO$_2$ amounts are found on working days, especially in urban areas. These high emissions are mainly due to road transport, which is the largest NOx contributor in Europe (Crippa et al., 2018) and emits up to 90% NO$_2$ in Madrid (Borge et al., 2014).

The ML-estimated emission strengths for Madrid are presented in Figure 4. High NO$_2$ emission sources on weekdays are evenly distributed around the city center (Figure 4a). However, for weekends, the northeastern regions close to the airport, far away from the city center, are the main sources, and no obvious sources are observed in the southwestern regions (Figure 4b). The total NO$_2$ emission strength in the urban area (dashed rectangles) during weekends (7.24×10$^{24}$ molec./s) are smaller than those observed during weekdays (9.85×10$^{24}$ molec./s) by about 26%. This result is similar to the result observed in another European city – Helsinki, where the weekly variability of traffic-related emissions was reduced by 30% at weekends (Ialongo et al. 2020). By subtracting weekends' emissions from the ones of weekdays (Figure 4c), we found that the dominant NO$_2$ sources are in east-to-northeast and south-to-southwest regions, where the residential areas and working places are mainly located (Figure A- 11). The orographic feature further causes the accumulation of NO$_2$ in these regions (see Section 3.2). The wind-assigned anomalies and correlation plots are presented in Figure A- 12. Note that slightly higher scattering in the results at weekends is mostly due to fewer data points.

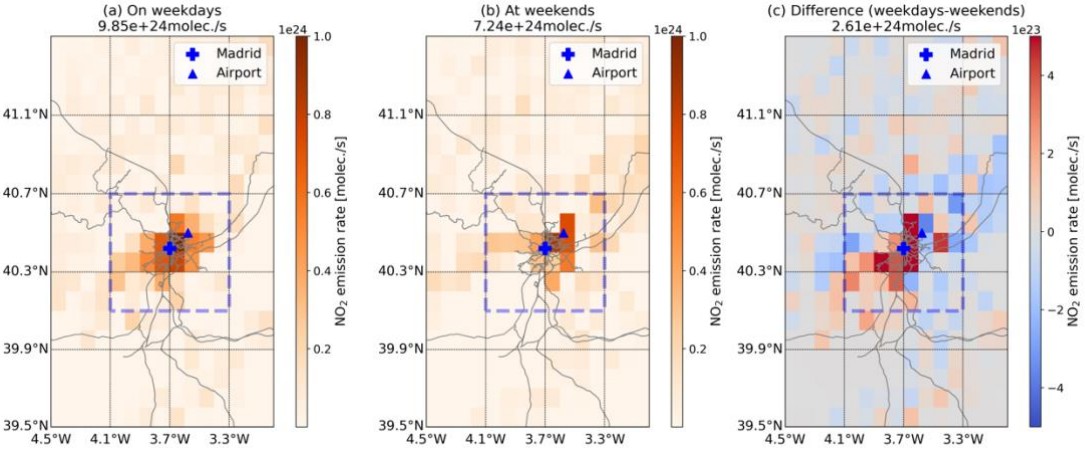

265

**Figure 4: same figures as Figure 3, but for the Madrid area. Number in each figure's title presents the total emission rate in the dashed rectangle (70 ×70 km$^2$).**

### 3.4 COVID-19 lockdown effect

The current global pandemic caused by coronavirus disease (COVID-19) largely impacts human life and the economic situation. To minimize the spread of the COVID-19 SARS-CoV-2 virus, countries around the world have enforced lockdown measures. Recent studies have reported decreasing NOx concentrations in the atmosphere due to lockdown, and additional reductions with more stringent lockdown, such as in Spain (Abdelsattar et al., 2021; Barré et al., 2021; Sun et al., 2021; Liu et al., 2021; Vîrghileanu et al., 2020; Keller et al., 2021; Bauwens et al., 2020; Fan et al., 2020; Huang and Sun, 2020). An approximate decrease of 40% in NO$_2$ is observed by OMI in Riyadh (Abdelsattar et al., 2021). Bauwens et al. (2020) illustrates the impact of COVID outbreak on NO$_2$ based on TROPOMI and OMI observations. The averaged NO$_2$ column decreases by ~29% derived from TROPOMI observations and by ~21% derived from OMI observations in Madrid during lockdown period (Bauwens et al., 2020). The NO$_2$ reductions are strongly related with the lockdown policy and is also presented in the study by Levelt et al. (2022) and it reports that NO$_2$ column amounts decreased by 14 % - 63 % in megacities globally. A sharp reduction of 54% in the NO$_2$ tropospheric column amounts was observed in Madrid during the lockdown period and 36% during the transition period. The time series of TROPOMI tropospheric NO$_2$ columns displays an obvious decrease since the lockdown started in early 2020 (Figure A- 13). The NO$_2$ amounts reach the lowest values in April 2020 and in the meanwhile they are gradually back to normal levels as in previous years. We analyze the same seasonal period in 2019 (before lockdown, March – June 2019) and in 2020 (during the lockdown, March – June 2020) for Riyadh and Madrid.

Figure 5 presents the spatial distribution of estimates before and during the lockdown in Riyadh. NO$_2$ emissions decreased by 21% from $1.24\times10^{26}$ molec./s before lockdown to $9.79\times10^{26}$ molec./s during lockdown. The spatial distribution of estimates during lockdown is similar to that at weekend, when significant decreases are observed along Highway 65 and emissions are generally reduced in the city center and in the areas where the cement plant and power plants are located.

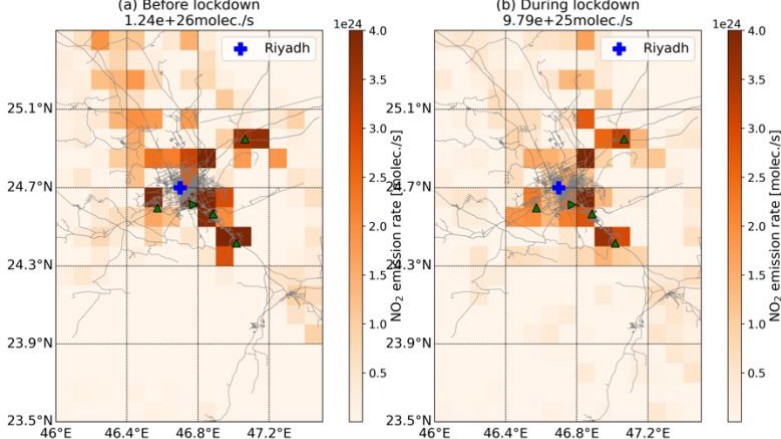

**Figure 5: Averaged ML estimated emission strengths before lockdown (March – June 2019) and during lockdown (March – June 2020). Number in each figure's title presents the total emission rate.**

The $NO_2$ emission estimate in the urban area of Madrid is about $1.04 \times 10^{25}$ molec./s before lockdown and it decreases by 60% to $4.04 \times 10^{24}$ molec./s during the lockdown period (Figure 6). This result fits well with the recent studies (Baldasano, 2020; Barré et al., 2021; Guevara et al., 2021). European Environment Agency (EEA) also reported a 56% - 72% reduction in $NO_2$ concentrations in Madrid based on in situ monitoring data (EEA report, 2020). Even compared to the emission at weekends, the lockdown emission was reduced by 43%. The regions with high $NO_2$ emissions are constrained only in the east of Madrid, where there are residential areas. Note that the lockdown spatial pattern reproduces that observed at weekends during the whole period (Figure 4b), corroborating that $NO_2$ emissions are highly related to transportation. Civil aviation was also restricted during the lockdown and thus less $NO_2$ emission strength is observed close to the airport. The reduction is larger than that in Riyadh as Madrid was under a very strict lockdown policy.

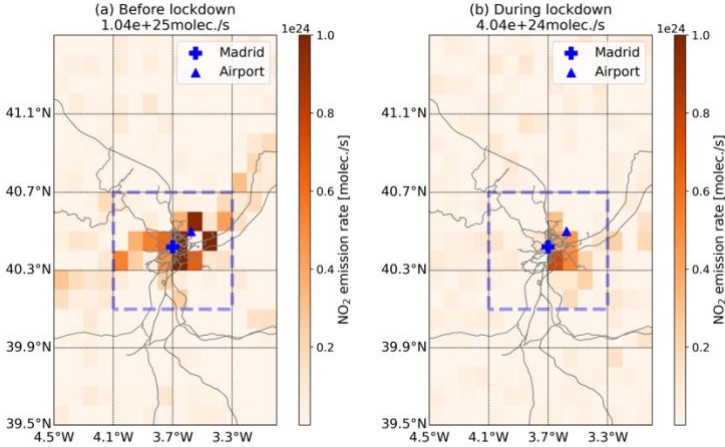

**Figure 6: same figures as Figure 5, but for the Madrid area. Number in each figure's title presents the total emission rate in the dashed rectangle (70 ×70 km²).**

## 4. Uncertainty analysis

### 4.1 Different choices for $\alpha$ and $\tau$

305    The angle ($\alpha$) of the emission cone is an empirical value, so as the lifetime/decay time ($\tau$) for $NO_2$. They can introduce uncertainties and thus, different values for $\alpha$ and $\tau$ are used to investigate their impacts on emissions. The spatial patterns of the estimates with using different $\alpha$ or $\tau$ are quite similar. The absolute values of emission rate increase with the increasing $\alpha$ (see Figure 7-left). A change of 10° in $\alpha$ introduces a difference of less than 3.2%. A decrease of 1.5% is observed when using $\alpha = 50º$, and an increase of 1.4% is observed for $\alpha = 70º$, as compared to $\alpha = 60º$. The increasing values of $\tau$ result in lower

310    estimates (see Figure 7-right). With respect to the result obtained with $\tau = 4h$, the estimate increases by ~42% for $\tau = 3h$, and it decreases by ~20% for $\tau = 5h$.

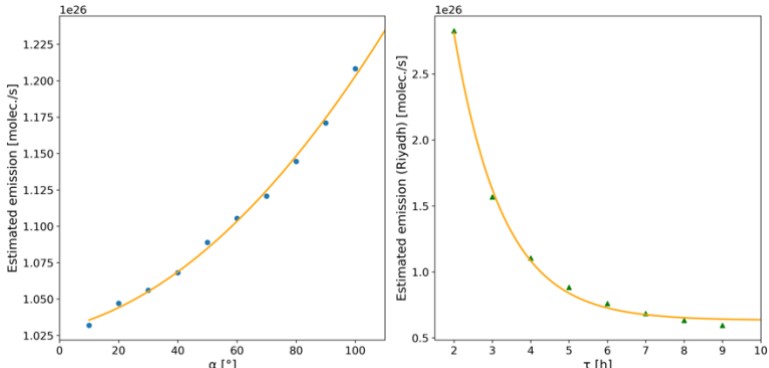

**Figure 7: Estimated emissions under different cone angle $\alpha$ (left) and $NO_2$ lifetime $\tau$ (right) based on TROPOMI data in Riyadh in 2019.**

315    ### 4.2 Different choice of wind field segmentation

The wind field segmentation is decided based on the predominant wind fields. We chose a different segmentation for Riyadh (i.e., SW: 45° - 225° and NE for the rest fields) and for Madrid (i.e., SE: 45° - 225 ° and NW for the rest fields). The spatial pattern of the estimates in Riyadh is similar with previous results (8a), whereas some unexpected positive emissions are obtained in southwest of Madrid. An increase of 12.5% in Riyadh and 8.6% in Madrid are estimated. Using different wind

320    segmentation leads to different spatial distributions of estimates, especially in Madrid where the topography (e.g., land cover, altitude) is more complicated than in Riyadh.

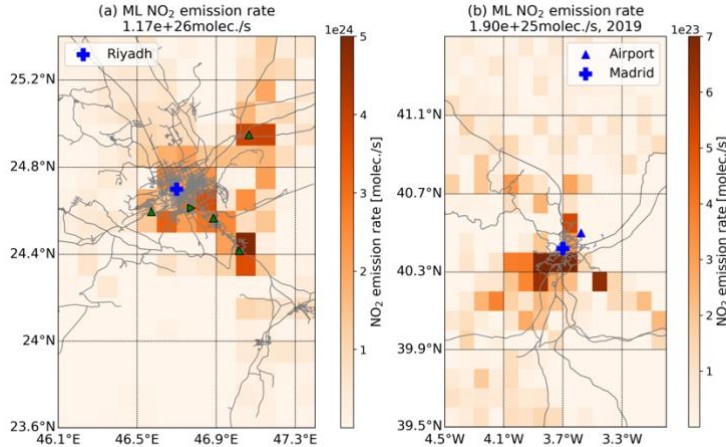

**Figure 8: (a) similar to Figure 1(d), but using SW-NE wind field segmentation; (b): similar to Figure 2(d), but using N-S wind field segmentation. Note that data are based on TROPOMI data in 2019.**

4.3 Different choice of wind field on vertical and horizontal dimension

The wind speed increases with altitude (Figure A- 2), whereas the distribution of wind directions stays similar. Approximate increases of 19% and 39% in wind speed at 100 m are observed in Riyadh and Madrid, respectively. The estimates change slightly in both cities, as the wind-assigned method compensate the increases on both wind field.

To limit the computational effort, we simplified the wind field on horizontal distribution to be evenly distributed, i.e., constant wind speed and wind direction over the study area at each time gap (1 hour). This might introduce some errors and thus, a full year of data in 2020 are used to investigate the uncertainty. The wind direction and speed are interpolated at each pixel center, as ERA5 wind is at a spatial resolution of 0.25º × 0.25º. Either the spatial distribution or the estimated emission is similar to those with constant wind field in both cities. The estimates change by 1.9% in Riyadh and by -1.3% in Madrid. The pixel-to-pixel difference on average is $6.8{\times}10^{21}$ ($\pm\ 4.6{\times}10^{23}$) molec./s in Riyadh and $-8.3{\times}10^{20}$ ($\pm\ 4.5{\times}10^{22}$) molec./s in Madrid.

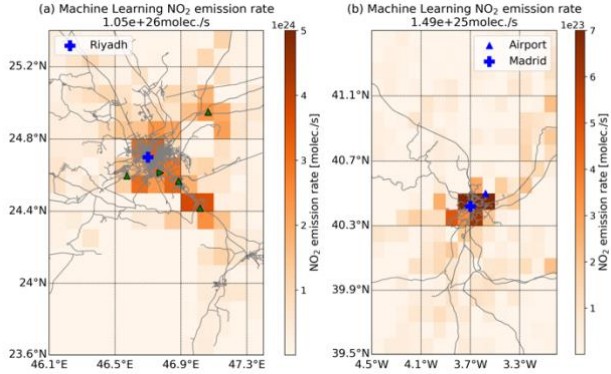

**Figure 9: Similar to Figure 5(c) and Figure 6(c) but using spatially varying wind field.**

## 5. Conclusions

The paper proposes a combination of wind-assigned anomalies and Machine Learning (ML) methods to estimate the average tropospheric $NO_2$ emission strengths and its spatial pattern derived from TROPOMI observations from April 2018 to June 2022. The ADAM algorithm, as one of the Gradient Descent algorithms in ML is chosen because of its high efficiency and little cost requirement.

Riyadh is first used as a test site due to its high population density, remote from other sources, and favorite weather conditions, which allow for the high availability of space-based observations. Consistent wind-assigned plumes are found based on the TROPOMI measurements, so as the ML-trained plumes. A very good correlation between them is obtained with an $R^2$ value of 1.0 and a slope of 0.99. The spatial pattern of the estimated emission strengths on the main sources near the city center agrees with the results from Beirle et al. (2019) as well. Several $NO_2$ emission hotspots, associated with the cement plant and power plants, are discernible. The total emission rate over the whole area is about $1.04 \times 10^{26}$ molec./s, which is higher than the previous study ($8.5 \times 10^{25}$ molec./s, Beirle et al., 2019). This difference might be due to the different study period and methods. These results suggest that our combined method works properly and is reliable.

We extended this method to the (mega)city of Madrid, Spain. The averaged $NO_2$ emission estimates are $1.80 \times 10^{25}$ molec./s in total and the dominant emitting area is around the city center, especially in the north-to-northeast and south-to-southeast regions. The region with the international Madrid-Barajas Airport in the northeast is also distinguished with high emission rates, as aircraft exhaust emissions are highly enriched in $NO_2$ during taxiing and taking off (Herndon et al., 2004). The orographic feature also causes $NO_2$ accumulation in the NE-SW regions, along the Guadarrama mountains range.

$NO_2$ emission is highly related to transportation and thus, $NO_2$ emission changes between weekdays and weekends are investigated as well. Different weekly cycles of $NO_2$ are observed in Riyadh and Madrid. The lowest $NO_2$ column abundances are observed on Fridays, followed by the ones on Saturdays in Riyadh. The $NO_2$ emissions were reduced by 24% at weekends, and high reductions are found near the city center and the areas along Highway 65. Regions in the west and southwest of Madrid are not main $NO_2$ emitting areas at weekends but are on weekdays, indicating that many working places are located in the southwest. The estimates are $9.85 \times 10^{24}$ molec./s on weekdays and $7.24 \times 10^{24}$ molec./s at weekends in the urban area (70 km × 70 km$^2$). This 26% reduction in $NO_2$ emission is mainly due to commuting from home to the city center and working places.

Many studies have demonstrated that the lockdown policy response to the COVID-19 pandemic reduces $NO_2$ emissions (Barré et al., 2021; Sun et al., 2021; Liu et al., 2021; Vîrghileanu et al., 2020; Keller et al., 2021; Bauwens et al., 2020; Fan et al., 2020; Huang and Sun, 2020). Countries like Spain imposed a very stringent lockdown since March 2020. An average reduction of 60% in $NO_2$ emissions is observed during lockdown (March – June 2020) compared to the period of March – June 2019. The regions with dominant $NO_2$ emissions during lockdown are limited in the east of Madrid where there are residential areas.

Reduced $NO_2$ emissions (27%) were observed in Riyadh, especially near the city center. This reduction is much smaller than that in Madrid, as the latter was under a very strict lockdown regulation.

Our easy-to-apply method has successfully probed its consistency and reliability in two contrasting examples (Riyadh and Madrid). However, application in some areas with complicated emission source distribution and topography might not be feasible. The varying decay time for short-lived species in different regions and seasons is another important factor affecting the estimates of emissions. We plan to include these refinements in future studies to reduce the uncertainties of both the wind-assigned anomaly method and the ML approach. The spatial distributions of estimates generally show checkerboard-like

structures. We assume that these structures indicate that the inversion attempts to resolve fine structure which is poorly constrained by the observation. When we converge to a stable solution with minimal bias, we are confident that spatially averaged retrieved emissions are more realistic. It is our hope that the method presented here can be applied to other key gases such as carbon dioxide or methane for which the background concentration needs to be considered, and in other regions. Meanwhile, the powerful ML framework might allow to investigate related questions, perhaps a joint estimation of $NO_2$

lifetime and emission strength would be possible.

*Data availability*. The TROPOMI data set is publicly available from https://scihub.copernicus.eu/ (last access: 18 January 2022; ESA, 2020). The access and use of any Copernicus Sentinel data available through the Copernicus Open Access Hub are governed by the legal notice on the use of Copernicus Sentinel Data and Service Information, which is given here:

https://sentinels.copernicus.eu/documents/247904/690755/Sentinel_Data_Legal_Notice (last access: 18 January 2022; European Commission, 2020).

*Author contributions*. Qiansi Tu, Frank Hase, Zihan Chen, and Matthias Schneider developed the research question. Qiansi Tu wrote the manuscript and performed the data analysis with input from Frank Hase, Zihan Chen, Matthias Schneider, Omaira

García, Farahnaz Khosrawi, Shuo Chen and Jason Cohen. All authors discussed the results and contributed to the final manuscript.

*Competing interests*. The authors declare that they have no conflict of interest.

*Acknowledgements*. We acknowledge Quan Ngoc Pham from Interactive Systems Lab, Karlsruhe Institute of Technology for the useful technical support in the ML model. We would like to thank Emissions of atmospheric Compounds and Compilation of Ancillary Data (ECCAD) for providing CAMS-REG-AP inventory data. Thanks should also go to the TROPOMI team for making $NO_2$ data publicly available. We also acknowledge the project of Joint R&D and Talents Program funded by the Qingdao Sino-German Institute of Intelligent Technologies (kh0100020213319) and the project of Transnational

Interoperability Rules and Solution Patterns in Collaborative Production Networks based on IDS and GAIA-X funded by Ministry of Science and Technology, PRC (SQ2021YFE010470).

**Appendix**

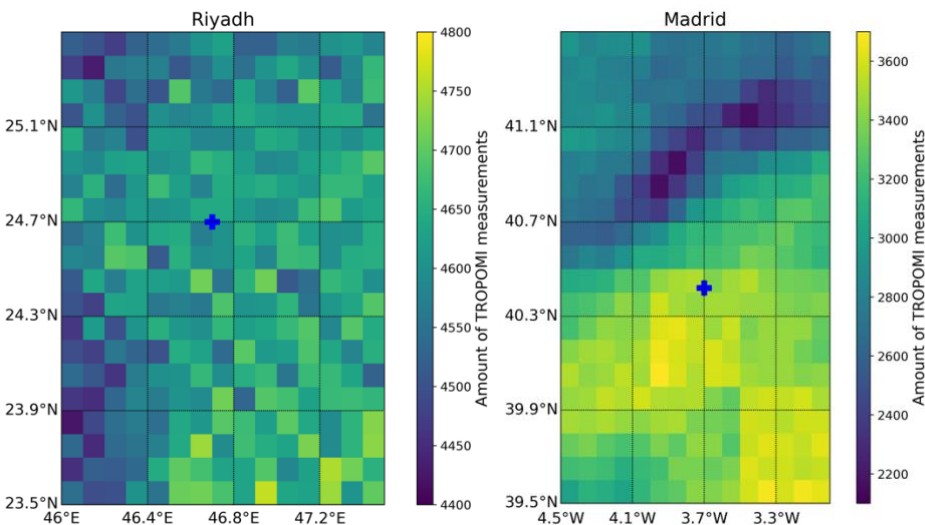

**Figure A- 1: Amount of TROPOMI measurements in each 0.1° grid pixel for Riyadh and Madrid during May 2018 – June 2022.**

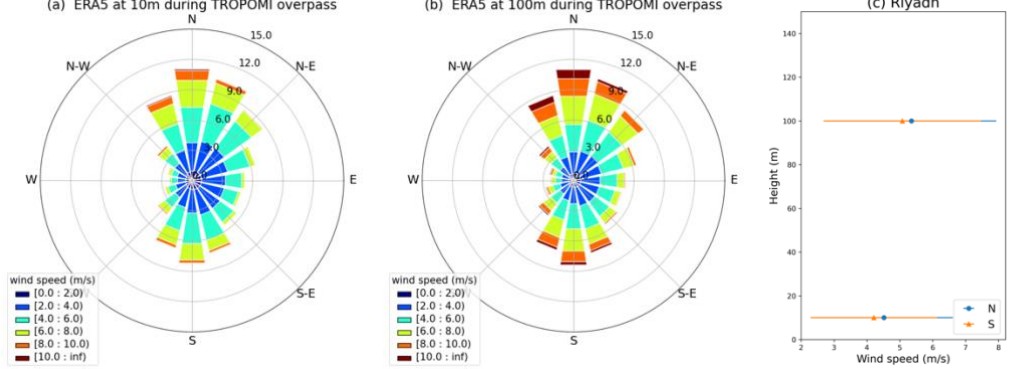

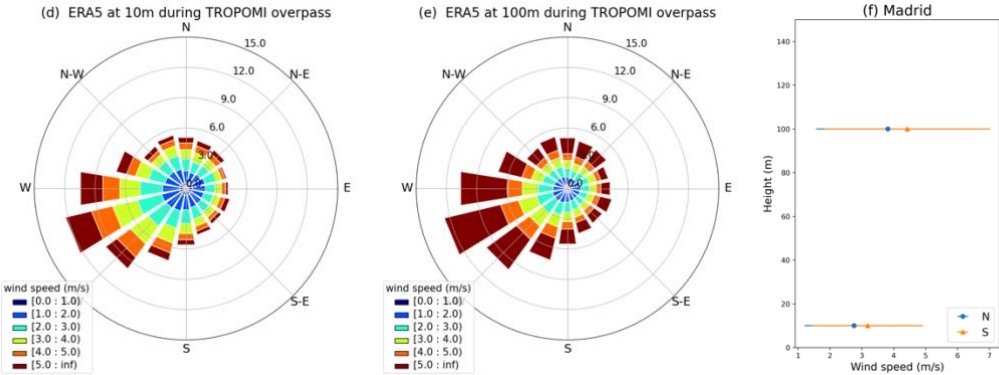

**Figure A- 2: Windrose of ERA5 model wind at 10 m and 100 m in the daytime during TROPOMI overpasses, and wind speed at two levels in Riyadh ((a)-(c)), and in Madrid ((d)-(f)).**

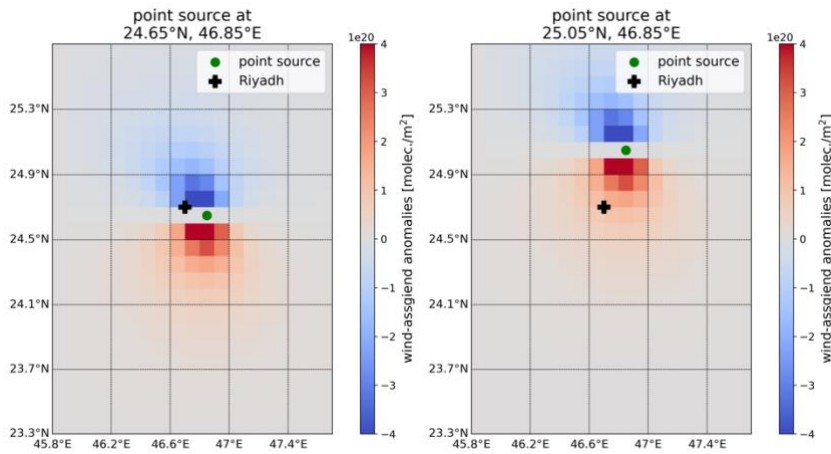

**Figure A- 3: Examples of wind-assigned plume for the point source at 24.65ºN, 46.85ºE and at 25.05ºN, 46.85ºE in Riyadh.**

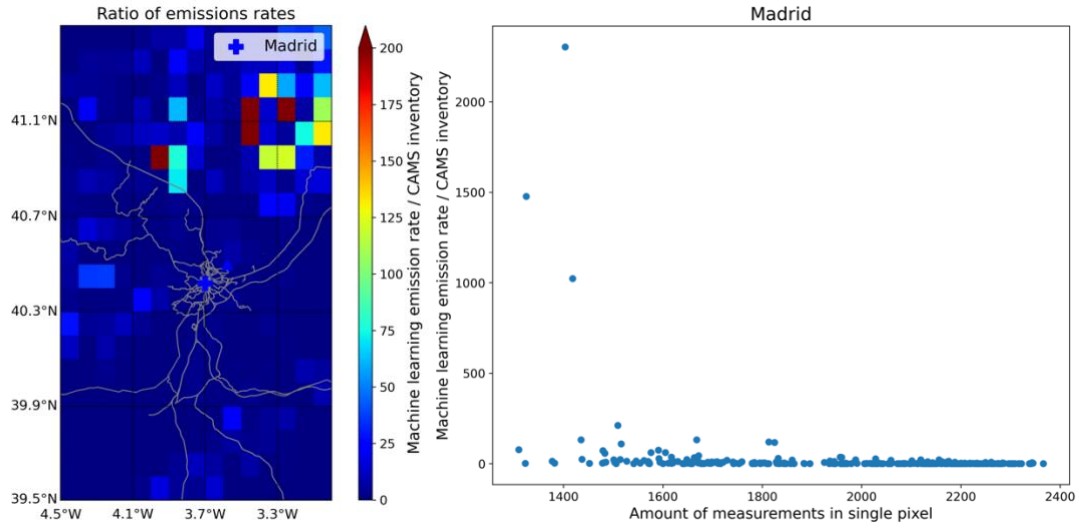

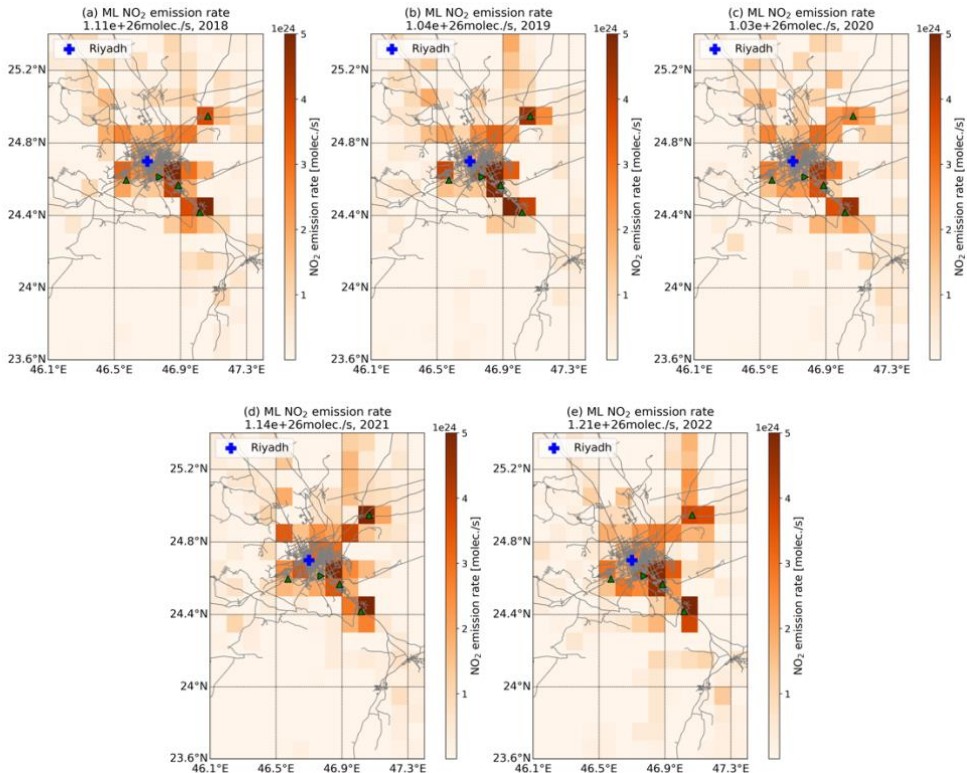

**Figure A- 4: left: Ratio of emission rates derived from Machine learning method and the CAMS inventory in 2020, right: correlation**
**between the amount of measurements in a single pixel and the ratio of emission rates.**

**Figure A- 5: yearly averaged estimated NO₂ emission rates in Riyadh for the years 2018 to 2022. Note that data in 2018 started from May and data in 2022 ended in June.**

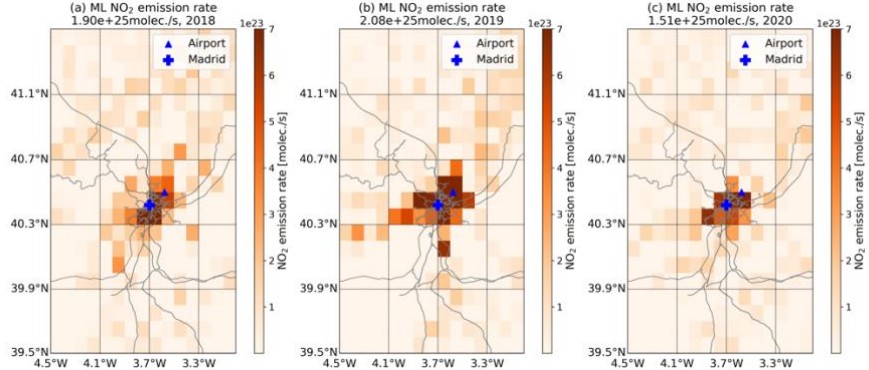

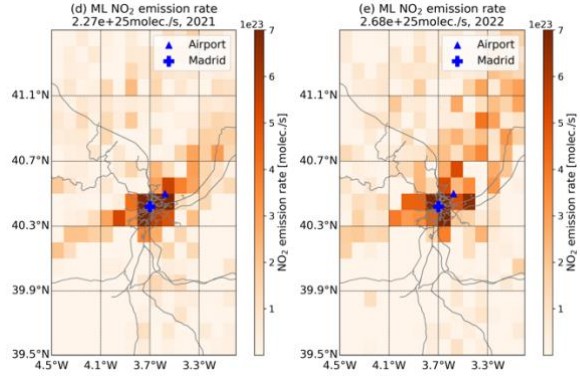

Figure A- 6: similar to Figure A- 5, but for the Madrid area.

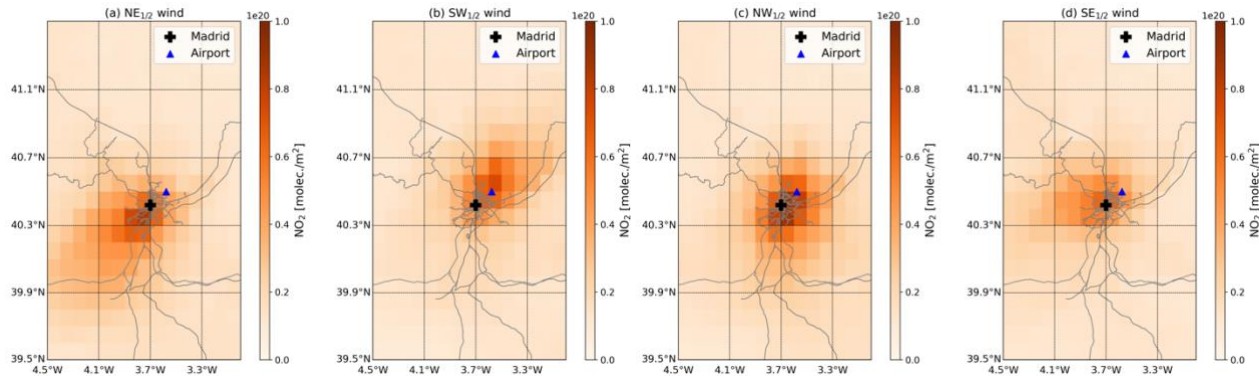

**Figure A- 7: TROPOMI tropospheric NO₂ column for narrow wind regimes covering (a) NE$_{1/2}$ (0º-90º), (b) SW$_{1/2}$ (180º-270º), (c) NW$_{1/2}$ (270º-360º), and (d) SE$_{1/2}$ (90º-180º), respectively.**

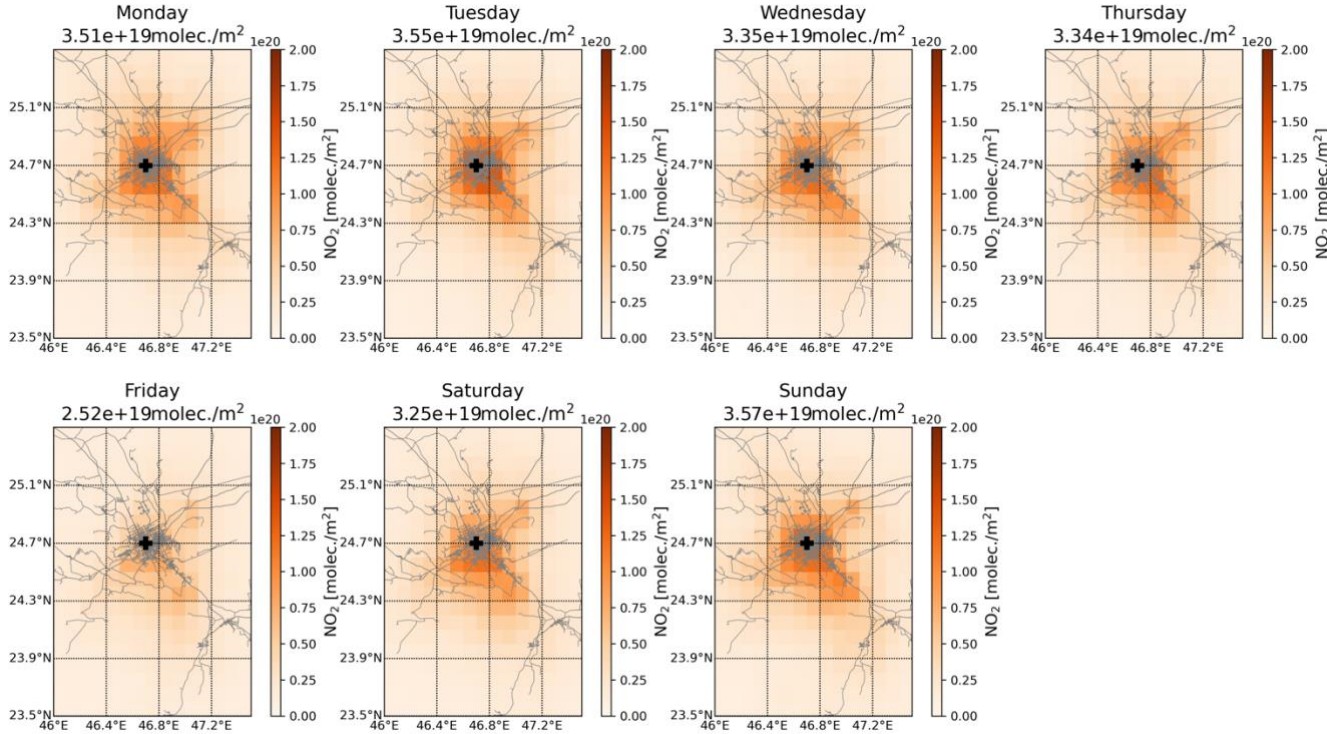

**Figure A- 8: TROPOMI tropospheric NO₂ column during week in Riyadh. Number in the figures' title represents the average column abundances over the area.**


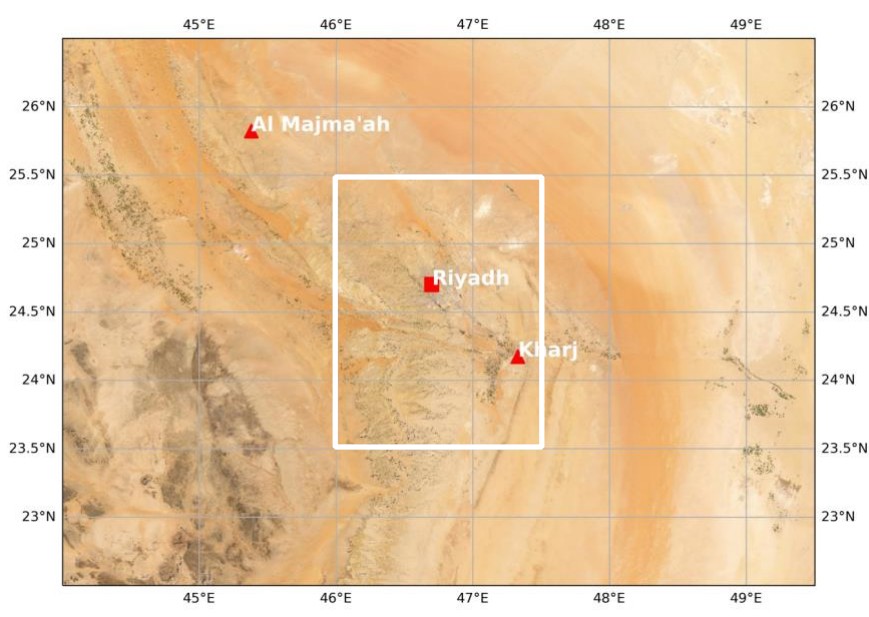

**Figure A- 9: Map for Riyadh. Area in the white rectangle represents the study area. © Esri**

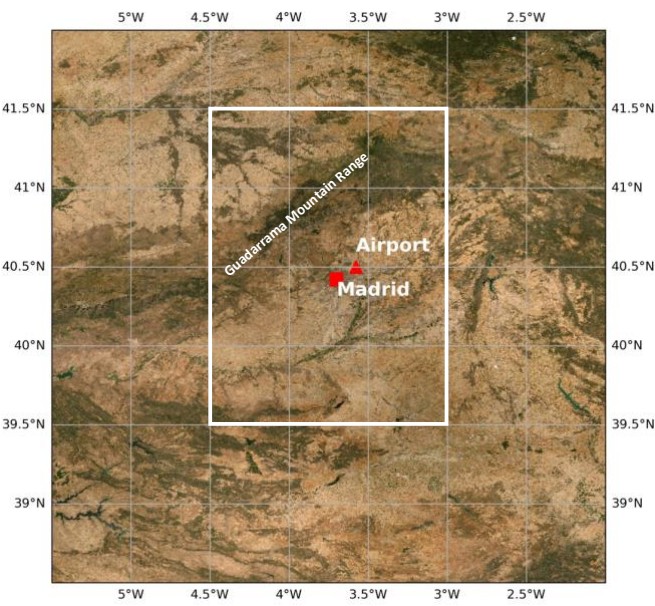

**Figure A- 10: same as Figure A- 8, but for the Madrid area.**

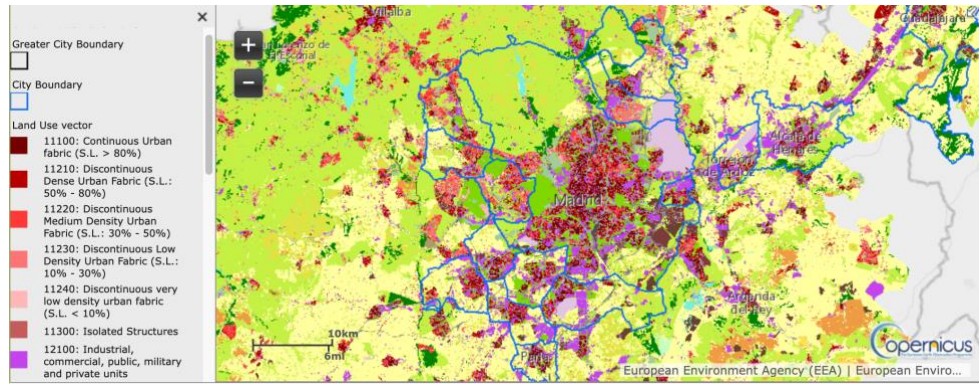


Figure A- 11: Map of the study area (up) © Esri and zoom version (bottom, https://land.copernicus.eu/local/urban-atlas/urban-atlas-2018, last access: 25 April 2022) for Madrid.

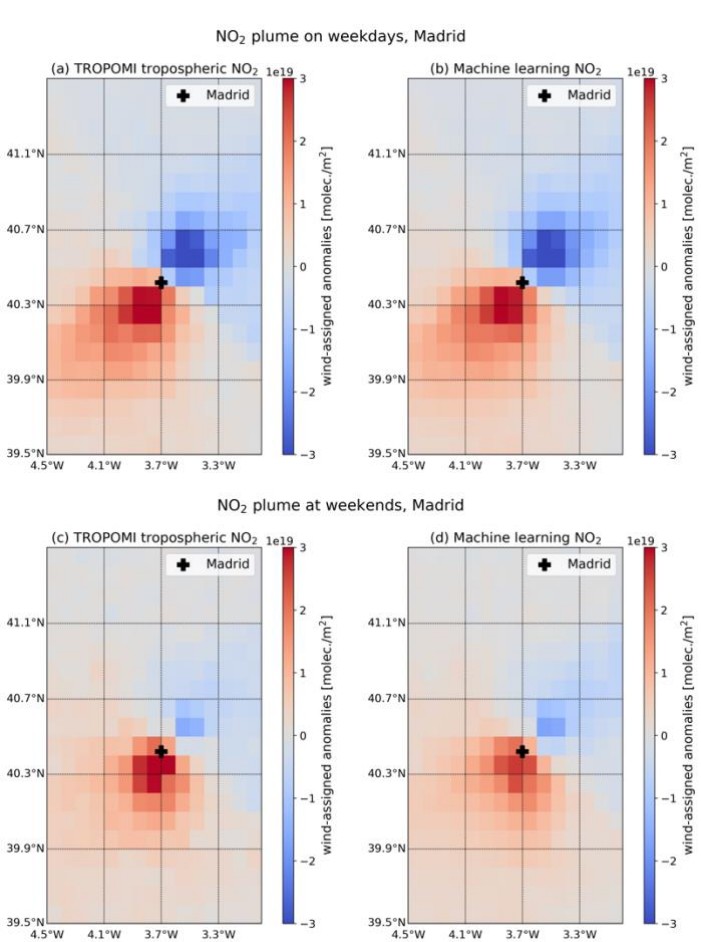


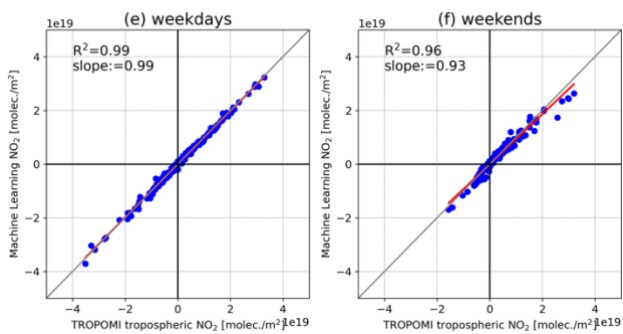

**Figure A- 12: Wind-assigned plumes derived from TROPOMI observations (a-b), ML method (c-d), and their correlation plots (e-f) on weekdays (left) and at weekends (right) in Madrid.**

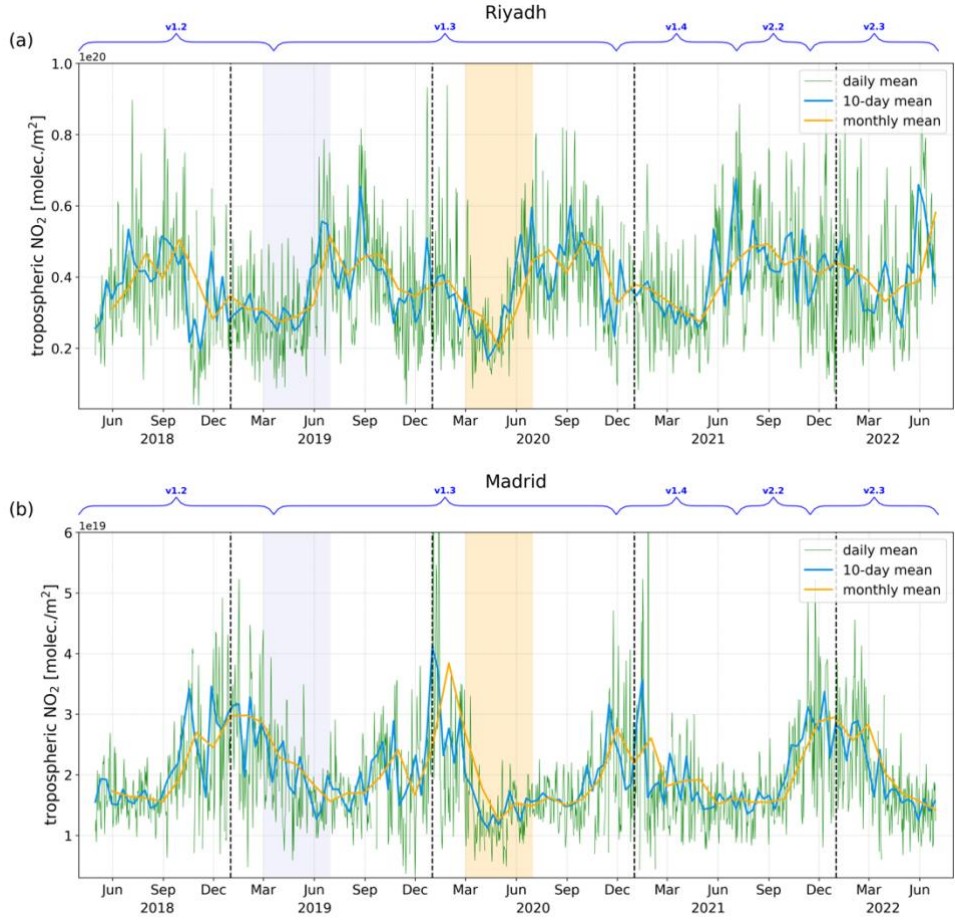


**Figure A- 13: Time series of TROPOMI tropospheric NO₂ columns in terms of daily, 10-day and monthly mean in (a) Riyadh and (b) Madrid. Areas marked with lavender and orange colors are the study periods in 2019 and 2020, respectively. The annotations on top of each figure represent the different versions of data sets.**

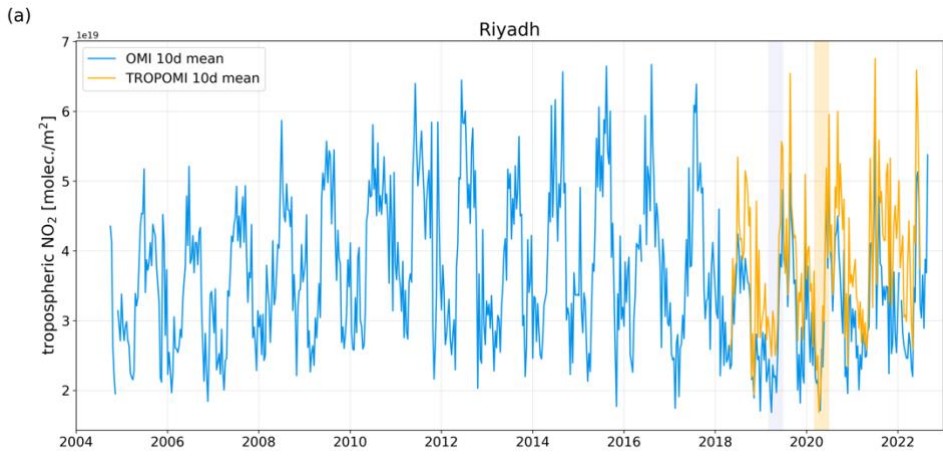


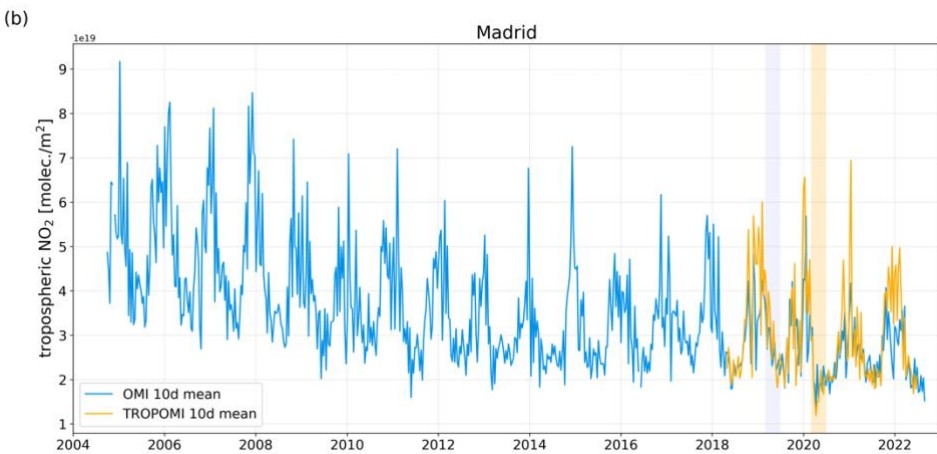

**Figure A- 14: Time series of TROPOMI and OMI tropospheric NO₂ columns in terms of 10-day mean in (a) Riyadh and (b) Madrid. Areas marked with lavender and orange colors are the study periods in 2019 and 2020, respectively**

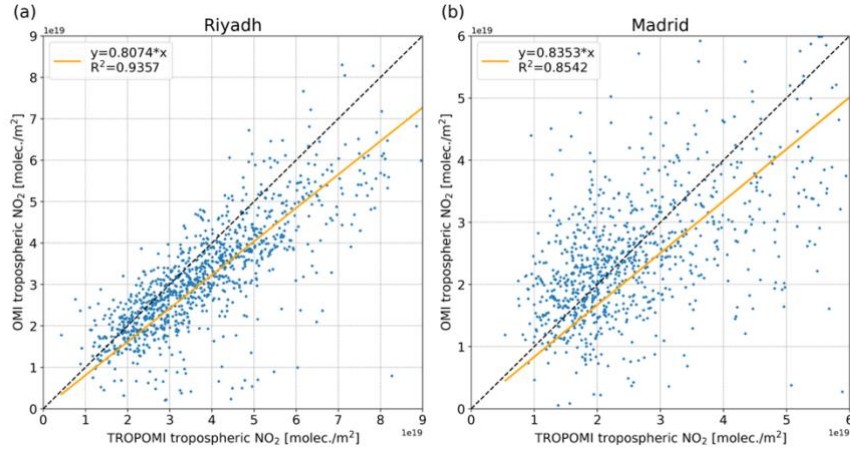

**Figure A- 15: Correlation plot between TROPOMI and OMI tropospheric NO₂ columns.**

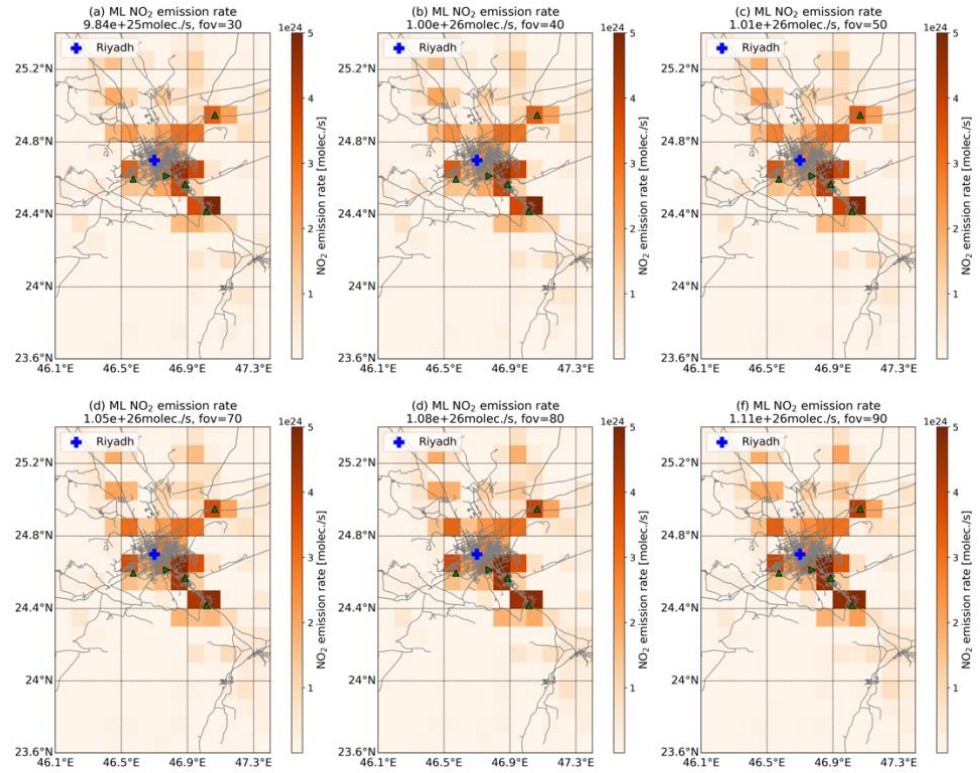

**Figure A- 16: Estimated emission rates in Riyadh for different angles ($\alpha$) of the emission cone from 30º to 90º.**

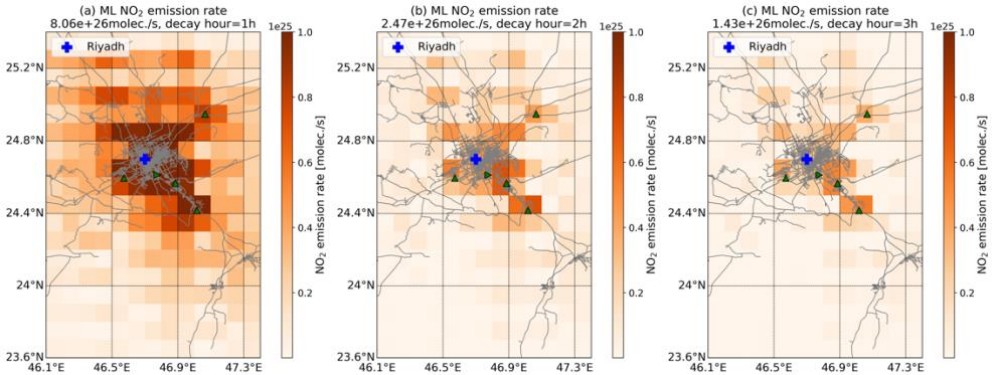

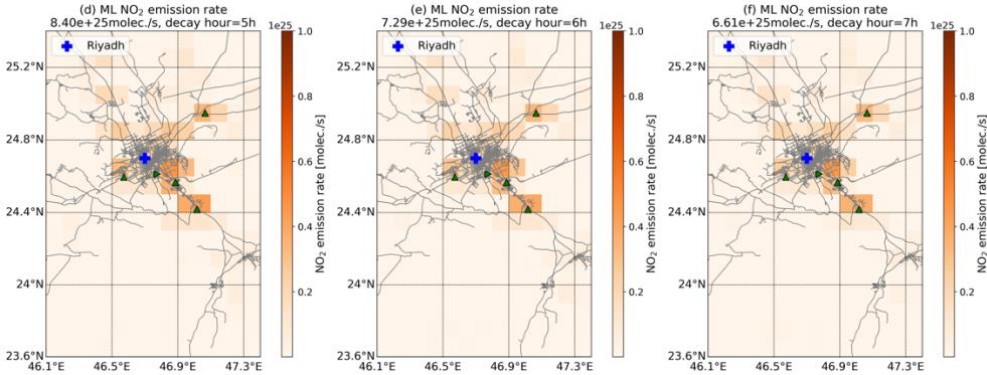

**Figure A- 17: Estimated emission rates in Riyadh for different decay hour ($\tau$) from 1h to 7h. Note that the colorbars is different than that in Figure 1(d) for covering larger range.**

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
