# Peer review of "Estimation of NO2 emission strengths over Riyadh and Madrid from space from a combination of wind-assigned anomalies and machine learning technique"

_Atmospheric Measurement Techniques, 2022_

## Author Comment (AC1)

**Response to Referee #1**

We would like to thank reviewer #1 for taking the time to review this manuscript and for providing valuable, constructive feedback and corresponding suggestions that helped us to further improve the manuscript.

In this authors' comment, all the points raised by the reviewer are copied here one by one and shown in blue color, along with the corresponding reply from the authors in black.

The paper "Estimation of NO2 emission strengths over Riyadh and Madrid from space from a combination of wind-assigned anomalies and machine learning technique" by Tu et al presents a new simple method to obtain the tropospheric NO2 emission strengths and their spatial patterns derived from the TROPOMI observations. It relies on wind-assigned anomalies and machine learning (ML) technique (the so-called Gradient Descent) and it is applied to 2 cases (Riyadh and Madrid), which have already been used for emission estimates by past literature, to which they compare to in a few words. They also present the weekend effect and the impact of the COVID at the 2 locations.

The paper is well written and easy to follow, the study is interesting and in the scope of the journal, but the method is only briefly described relying a lot on references, and only describing the technical implementation, and no error estimation is mentioned.

I recommend publication after some revision, including some more discussion and some testing cases (some are suggested below) to provide an estimation of the uncertainty of the method.

Specific comments:

----------------------------The paper to my point of view lacks some discussion about the importance of the choice of different parameters, that here have been fixed "once for all", as coming from a reference (sometimes on a quite different topic), e.g. the choice of the wind (line 85) is the ERA5 at 330m coming from an "empirical choice based on Tu et al (2022b) ...", but where the focus is methane. In the discussion of the Riyadh results (sect 3.1) the output value is compared to Beirle et al. 2019, using a different wind source (ECMWF) and height (450m) and no discussion at all is made to help the reader knowing if these choices have a (large) impact or not. Similarly, when discussing the differences with previous results from Beirle 2011 and 2019 only a sentence "The difference might be due to the different study periods and methods used." (line 162) is mentioned. Also for the Madrid case (line 184: "This discrepancy is partly due to the different periods, methods, and data sets used."). The author could make some tests on either different heights, or different wind source and give an estimation of the final outcome value. To estimate the impact of the selected choices, I would suggest the following tests:

1. - use 1 year of S5p data instead of 3. Has this a big impact? Is this helping the coherence with the Madrid results, where yearly inventories are available?

The data set has been updated to June 2022, i.e., the study period is May 2018 – June 2022. The yearly averaged estimates in Riyadh and in the Madrid area are presented in Figure A- 5 and A- 6 in the updated manuscript, respectively.

For Riyadh the spatial patterns of emissions are generally similar in the five years. Small changes in the NO2 emission are observed from 2018 to 2019. The yearly-average NO2 concentrations in 2019 and 2020 are

similar, which are $3.78 \times 10^{19} \pm 5.42 \times 10^{19}$ molec./s and $3.81 \times 10^{19} \pm 5.51 \times 10^{19}$ molec./s, respectively. The total emission rates are similar in 2019 and 2020 as well. That is to say, the amount of emission decreased during the COVID lockdown in early 2020 is compensated by higher emissions later in the same year. The estimate increases by 11% in 2021, as there is no lockdown in 2021. A small increase of 6% is observed in 2022. Note that results in 2019-2021 are based on TROPOMI observations over an entire year.

[Figure]

Figure A- 5: yearly averaged estimated $NO_2$ emission rates in Riyadh for the years 2018 to 2022. Note that data in 2018 started from May and data in 2022 ended in June.

A yearly decrease of emission rates is estimated to be 27% in Madrid area in 2020, compared to the value in 2019. This observation fits to the fact that a very strict lockdown was implemented in Madrid. The paler color in 2020 represents lower emissions mainly due to the COVID lockdown in spring, especially in the city center. Compared to the result in 2019, the spatial pattern in 2018 is slightly different. This is because the data in 2018 started from May and thus, the low $NO_2$ concentrations observed in the early year are not included. Higher emissions are observed in the northeast of Madrid city in 2022 compared with those in other years, because the wind direction tends to be more along southwest to northeast in 2022 (Figure R1).

[Figure]

Figure A-6: similar to Figure A-5, but for the Madrid area.

[Figure]

Figure R1: Windrose of ERA5 model wind at 10m for different years in the daytime during TROPOMI overpasses in Madrid.

2. - test another wind height (why not the surface? as the NO2 is a short-live species, the NO2 will follow the orography)

The referee is right that NO$_2$ is a short-lived species. It might introduce uncertainty that we referred the wind field at 975 hpa to be at 330 m, as the altitude varies in the study areas. To eliminate the error due to the varying surface height, we thus, decide to use the wind at 10 m (above surface) to estimate its emissions, and use the wind at 100 m as an uncertainty analysis. The corresponding results in the manuscript have been updated, and the discussion of the uncertainty from wind height has been added to section 4.3.

3. - what is the impact of S5p in term of pixel resolution (ie wrt to OMI larger pixels used in Beirle 2011 for Madrid)? Could a test be made by resampling S5p to OMI resolution and seeing the impact on the outcome emission estimate?

Thank you for this comment. The way how to bin/regrid the data might introduce an uncertainty. The whole year data in 2019 is used to investigate the uncertainty because it was not affected by the COVID lockdowns (see Figure R2 and Figure R3). When using a lower spatial resolution of 0.25°×0.25°, though the distinguished high emissions are smeared and merged into one pixel, the patterns are generally similar between lower and higher resolution. More information about emissions can be observed when using a higher spatial resolution (finer grids). The estimate binned in 0.25°×0.25° decreased by 13% compared to that using higher spatial resolution (0.1°×0.1°), whereas small changes (6%) are observed for data binned in 0.05°×0.05° in Riyadh. The total estimated emission rate changes slightly when changing the binned grid from 0.25°×0.25° to 0.1°×0.1° in the Madrid area (7%), and the total estimate increased by about 9% for data binned in 0.05°×0.05°.

[Figure]

Figure R2: (a)-(c) estimated emission rate in Riyadh and (d)-(f) averaged TROPOMI tropospheric *NO$_2$* in 2019. Data is binned in different spatial resolution, (a) and (d): 0.25°×0.25°, (b) and (e): 0.1°×0.1°, (c) and (f): 0.05°×0.05°.

[Figure]

Figure R3: Similar to Figure R2, but for Madrid.

We obtained OMI data with a spatial resolution of 0.25°×0.25°. The estimated emission rate in 2019 is $7.52\times10^{25}$ molec./s for Riyadh which is ~17% lower than that retrieved from TROPOMI observations. The total emission is $1.58\times10^{25}$ molec./s for Madrid area which is ~18% less than that retrieved from TROPOMI observations. The spatial patterns of estimates show some differences between TROPOMI and OMI, mainly due to their discrepancy in observations (Figure R2(a) and (d), Figure R3(a) and (d), and Figure R4). OMI data shows a better correlation with TROPOMI data for Riyadh ($R^2 = 0.95$) as compared to Madrid ($R^2 = 0.87$), see Figure R5. The yearly average $NO_2$ amounts are ~18% lower for OMI for Riyadh. For Madrid, TROPOMI data are ~18% higher near the city center than OMI, whereas slightly lower values are observed in the rural areas compared to OMI data. The biases between OMI and TROPOMI fits well with our estimates as mentioned above.

[Figure]

Figure R4: Averaged ML estimated emission strengths ((a)-(b)) obtained from OMI observations and OMI tropospheric $NO_2$ ((c)-(d)) in Riyadh and Madrid in 2019.

[Figure]

Figure R5: Correlation plot between TROPOMI and OMI tropospheric $NO_2$ columns in 2019 in Riyadh and in Madrid center.

4.  - be careful that the S5p NO2 dataset is an aggregate of different versions (010202, 0103xx since 20/3/2019 and 0104xx since 29/11/2020, see ROCVR here: https://mpcvdaf.tropomi.eu/index.php/nitrogen-dioxide). This should be mentioned in Sect 2.1 (v1.4 has an important change in the FRESCO cloud algorithm leading to larger NO2 columns, see e.g. Van Geffen et al., 2022) and a test on the impact of the change of version could also be interesting (as the

remote and urban NO2 columns changes differently, is this leasing to a different spatial result of the emissions?). The different versions should be added in figure A9 with the S5p NO2 time-series.

Thanks to the referee for this information, the text below has been added to the manuscript:

"The NO₂ data used in this study are obtained from the Sentinel-5P Pre-Operations Data Hub (https://s5phub.copernicus.eu/dhus/#/home), which provides level 2 datasets with three different data streams: the Non-Time Critical or Offline (OFFL), the Reprocessing (RPRO) and the near-real-time (NRTI) streams. The NRTI is available within 3 h after the actual satellite measurement and may sometimes be incomplete and has a slightly lower data quality (http://www.tropomi.eu/data-products/level-2-products, last access: 14 September 2022), and thus, this data set is not considered here. The RPRO data covers a time range of 30 April 2018 – 17 October 2018 and the OFFL data covers the remaining time period. Meanwhile, the NO₂ dataset is an aggregate of different versions. The RPRO data is v1.2, while OFFL includes several versions: v1.2 until March 20, 2019, v1.3 until 29 November 2020, v1.4 until 5 July 2021, v2.2 until 15 November 2021, and v2.3 until 17 July 2022 and v2.4 afterwards. An improved FRESCO cloud retrieval has been introduced in v1.4, which leads to higher tropospheric NO₂ columns over areas with pollution sources under small cloud coverage (van Geffen et al., 2022)."

Figure A9 (now is Figure A13 in the manuscript) has been updated according to the referee's comment. The study period is extended to June 2022.

[Figure]

Figure A-13: Time series of TROPOMI tropospheric NO₂ columns in terms of daily, 10-day and monthly mean in (a) Riyadh and (b) Madrid. Areas marked with lavender and orange colors are the study periods in 2019 and 2020, respectively. The annotations on top of each figure represent the different versions of data sets.

To study the impact of the version change, our method is applied to different datasets. To have the same study period, we chose v1.3 during December 2019 – June 2020, and v1.4 during December 2020 – June 2021 and v2.3 during December 2021 – June 2022. The TROPOMI tropospheric NO₂ column and estimated emissions in both cities are shown below. The mean tropospheric NO₂ concentrations for different period (i.e., corresponding to different version of data) are $3.4\times10^{19}$, $3.5\times10^{19}$, $4.2\times10^{19}$ molec./m$^2$ and the estimated emission rates are $8.95\times10^{25}$, $1.01\times10^{26}$, $1.21\times10^{26}$ molec./s, respectively in Riyadh. The values are $2.2\times10^{19}$, $1.8\times10^{19}$ and $2.2\times10^{19}$ molec./m$^2$ for the tropospheric NO₂ concentrations and $2.23\times10^{25}$, $1.60\times10^{25}$, $2.95\times10^{25}$ molec./s for the estimates, respectively in the Madrid area. The amounts of NO₂ concentration are different for different versions in different years. The wind is a very important component influencing the estimated emissions and the wind conditions are different in different years. Thus, it is hard to say whether the change of version leads to different spatial results of emission in different time periods.

[Figure]

Figure R6: Tropospheric NO₂ column (top) and estimates (bottom) derived from TROPOMI data in different version in Riyadh.

[Figure]

Figure R7: Similar to Figure R6 but in Madrid area.

Other minor comments:

5. About the COVID impact, the illustrations are interesting, but the context of other studies could be done better. Some studies explicitly mention Madrid, in maps or tables (eg Beuwens et al 2020, table 1 and Levelt et al., 2021, figure 3) and could be discussed.

The discussion is added to Section 3.4 according to the referee's comment:

"An approximate decrease of 40% in $NO_2$ is observed by OMI in Riyadh (Abdelsattar et al., 2021). Bauwens et al. (2020) illustrates the impact of COVID outbreak on $NO_2$ based on TROPOMI and OMI observations. The averaged $NO_2$ column decreases by ~29% derived from TROPOMI observations and by ~21% derived from OMI observations in Madrid during lockdown period (Bauwens et al., 2020). The $NO_2$ reductions are strongly related with the lockdown policy and is also presented in the study by Levelt et al. (2022) and it reports that $NO_2$ column amounts decreased by 14% - 63% in megacities globally. A sharp reduction of 54% in the $NO_2$ tropospheric column amounts was observed in Madrid during the lockdown period and 36% during the transition period."

6. Some of the figures in the annexe are "quick and dirty" (or give this feeling at least). They seem like simple print-screens, without any latitude & longitude coordiates, etcc (figures A5, A7).

Thanks for pointing it out. Figure A5 (now is A9) and A7 (now is A11) are updated as the referee recommended:

[Figure]

7. - line 33: "Though our analysis is limited to two cities as testing examples, the method has proved to provide reliable and consistent results." --> what are the errors and limitations of the method? this is not presented in the manuscript!

Thanks for raising up this important information. We have added another section about uncertainty analysis.

The limitations of the method are explained in Q8.

8. - line 34 (and in the conclusions): "Therefore, it is expected to be suitable for other trace gases and other target regions." --> be a bit less optimistic, here only the 2 "easier" cases have been presented, but there can be challenges for other places/gases.

Thanks. This sentence is modified to:

"Though our analysis is limited to two cities as testing examples, the method has proved to provide reliable and consistent results. It is expected to be suitable for other trace gases and other target regions. However, it might become challenging in some areas with complicated emission sources and topography, and specific $NO_2$ decay times in different regions and seasons should be taken into account. These impacting factors should be considered in the future model to further reduce the uncertainty budget."

So as the last paragraph in conclusion:

"Our easy-to-apply method has successfully probed its consistency and reliability in two contrasting examples (Riyadh and Madrid). However, application in some areas with complicated emission source distribution and topography might not be feasible. The varying decay time for short-lived species in different regions and seasons is another important factor affecting the estimates of emissions. We plan to include these refinements in future studies to reduce the uncertainties of both the wind-assigned anomaly method and the ML approach. The spatial distributions of estimates generally show checkerboard-like structures. We assume that these structures indicate that the inversion attempts to resolve fine structure which is poorly constrained by the observation. When we converge to a stable solution with minimal bias, we are confident that spatially averaged retrieved emissions are more realistic. It is our hope that the method presented here can be applied to other key gases such as carbon dioxide or methane for which the

background concentration needs to be considered, and in other regions. Meanwhile, the powerful ML framework might allow to investigate related questions, perhaps a joint estimation of NO$_2$ lifetime and emission strength would be possible."

9. - Sect 2.1: give more details on the TROPOMI NO2 data used. Which version? OFFL or a reprocessed? which version number? (at least mention that different version exist and give references)

Thanks. Please see our reply for Q4.

10. - lines 85-88: see comment above about wind selection. Methane has a much longer lifetime than NO2 (about 10 years vs a few hours), so explain why the choices made for methane are still ok for NO2 or find some estimation of the uncertainty related to this choice.

We have added discussion about the uncertainty of the choice of wind field (on vertical and horizontal dimension and segmentation) in Section 4.

11. - line 96: how much impact has the choices of tau (4h for Riyadh and 7 for Madrid) on the result? what are the ideas to estimate this value for other trace gases or other target regions? (to follow my comment on the 2nd bullet here above!) - lines 130-133: I don't understand very well the scope of increasing the area, and then removing the "outmost ring" (ring is a bit misleading as the ROI is a rectangle)

The choices of $\tau$ are adopted from the studies of Beirle et al. (2011, 2019). We discussed the uncertainty of the choices of $\tau$ so as the $\alpha$ in the additional section 4:

"The angle ($\alpha$) of the emission cone is an empirical value, so as the lifetime/decay time ($\tau$) for NO$_2$. They can introduce uncertainties and thus, different values for $\alpha$ and $\tau$ are used to investigate their impacts on emissions. The spatial patterns of the estimates with using different $\alpha$ or $\tau$ are quite similar. The absolute values of emission rate increase with the increasing $\alpha$ (see **Error! Reference source not found.**-left). A change of 10º in $\alpha$ introduces a difference of less than 3.2%. A decrease of 1.5% is observed when using $\alpha = 50º$, and an increase of 1.4% is observed for $\alpha = 70º$, as compared to $\alpha = 60º$. The increasing values of $\tau$ result in lower estimates (see **Error! Reference source not found.**-right). With respect to the result obtained with $\tau = 4h$, the estimate increases by ~42% for $\tau = 3h$, and it decreases by ~20% for $\tau = 5h$.

[Figure]

Figure 7: Estimated emissions under different cone angle α (left) and NO$_2$ lifetime τ (right) based on TROPOMI data in Riyadh in 2019."

The outermost rectangle of target area can be affected by the further outer area. Thus, we extend the area with 2-grid width. To avoid misleading, the "outmost ring" is changed to "outmost rectangle within 2-grid width".

12. - lines 141-145: although this paragraph is given to explain things, although for a nonexpert on ML as myself, it is just creating confusion, with a lot of different names (without explanations or references), to end up with "This practice is not necessary with our problem." --> why?

It seems this paragraph is not very relevant with our work and might make readers confused. Therefore, we decide to remove it.

13. - line 157: suggestion to "The estimated emission strengths based on the ML model *(Fig. 1d)* show a very similar spatial pattern to the results in Beirle et al. (2019) (Fig. 2)."

Thanks. Changed accordingly.

14. - line 163: see comment above about discussing impact of different choices and tests that could be made to estimate uncertainty of the present method.

Thanks for this important comment. Another section of uncertainty analysis (including different choice of α, τ, wind field segmentation and wind on vertically and horizontal dimension) has been added to the manuscript.

15. - line 184: "with a spatial resolution of 0.05° × 0.1° - 0.1° × 0.1° in longitude and latitude" --> I don't understand this resolution notation.

CAMS-REG-AP emission inventory is developed for the European domain at a 0.05° × 0.1° grid resolution (Kuenen et al., 2022). Meanwhile, it also provides a resolution of 0.1° × 0.1°.

16. "on a yearly basis" --> what year is considered in CAMS-REG-AP? or, if different years are available, can a test be made with only one year of TROPOMI to see if the year-to-year variability is similar?

We used the CAMS-REG-AP v4.2, which was the latest version covering 2000-2017 (Kuenen et al., 2022) when we prepared the manuscript. The inventory in 2017 was then used in the study.

We realized that an updated data version is available (v5.1 for 2000-2018, v4.2-ry for 2018-2019 and v5.1-BAU2020 for 2020), and the time period is extended to 2020 (https://eccad3.sedoo.fr/#CAMS-REG-AP). The spatial resolution of v5.1-BAU2020 is 0.05°×0.1° on latitude and longitude and this data set is regridded to 0.1° × 0.1° to be consistent with the resolution in previous data sets. Note that the impacts related to Covid-19 lockdowns are neglected in this version.

The time series of the inventory is presented in Figure R8 below and a significant decrease is observed over time. A nearly 18% reduction is found in 2020 compared with that in previous year, which is close to the expectation of the Plan A. Figure R9 presents the spatial distribution of CAMS inventory in 2018 - 2020. The spatial patterns of emissions are similar in different years, except slightly decreases in the city area in 2020.

[Figure]

Figure R8: Time series of total emission rate in Madrid area according to the CAMS-REG-AP inventory.

Figure R9: spatial distribution of CAMS-REG-AP inventory in 2018 - 2020.

Kuenen, J., Dellaert, S., Visschedijk, A., Jalkanen, J.-P., Super, I., and Denier van der Gon, H.: CAMS-REG-v4: a state-of-the-art high-resolution European emission inventory for air quality modelling, Earth Syst. Sci. Data, 14, 491–515, https://doi.org/10.5194/essd-14-491-2022, 2022.

Kuenen, J., Dellaert, S., Visschedijk, A., Jalkanen, J.-P., Super, I. and Denier van der Gon, H.2021Copernicus Atmosphere Monitoring Service regional emissions version 5.1 business-as-usual 2020 (CAMS-REG-v5.1 BAU 2020)Copernicus Atmosphere Monitoring Service [publisher],ECCAD [distributor],2021 doi.org/10.24380/eptm-kn40

17. - line 183: comment on the possible OMI vs TROPOMI resolution impact (see proposed test above). An error estimation would really help disentagle choices made with this approach and impact of time-periods (2005-2009 vs 2018-2021). How are the trends in NO2 around Madrid ? (see eg Georgoulias et al 2019 Fig 5)

The discussion of OMI data is added in the manuscript:

"The time series of tropospheric $NO_2$ observed by OMI since 2004 and by TROPOMI since 2018 in two study sites are shown in Figure A- 14 and their correlations are shown in Figure A- 15. $NO_2$ amounts increased since 2004 and reached highest value around 2016, except a sudden drop in 2013 in Riyadh. A continuous decrease is observed in Madrid and the COVID lockdown leads to a reduction of $NO_2$ emission in 2020. $NO_2$ concentration retrieved from the OMI observations are generally lower (slope=0.8074) than TROPOMI results with a mean bias of $6.3\times10^{18} \pm 9.8\times10^{18}$ molec./$m^2$ in Riyadh. The $R^2$ value in Madrid area ($R^2$=0.8542) is slightly smaller than the value in Riyadh ($R^2$=0.9357). However, the mean bias is less and the standard deviation is higher in Madrid area with a value of $1.9\times10^{18} \pm 1.2\times10^{19}$ molec./m2 (slope=0.8353). The ML emission rate retrieved from OMI observations (binned in 0.25°×0.25°) is 17% lower in Riyadh and 18% lower in Madrid area than those from TROPOMI observations. Thus, the discrepancy between this and previous study is mainly due to the data sets used."

[Figure]

Figure A- 14: Time series of TROPOMI and OMI tropospheric NO$_2$ columns in terms of 10-day mean in (a) Riyadh and (b) Madrid center. Areas marked with lavender and orange colors are the study periods in 2019 and 2020, respectively

[Figure]

Figure A- 15: Correlation plot between TROPOMI and OMI tropospheric NO$_2$ columns.

18. - line 190: what do you mean by "With an expectation, these actions may help to decrease NO2 concentrations by ~25% in the central area by 2020." ? We are in 2022!

This was assumed by Plan A issued in 2017. We modified the sentence to:

"In this context, Madrid City Council launched the Air Quality and Climate Change Plan for the city of Madrid (Plan A) in 2017, aiming at reducing pollution and adapting to climate change and ~25% reduction of $NO_2$ concentrations in the central area were assumed by 2020 (https://www.madrid.es/UnidadesDescentralizadas/Sostenibilidad/CalidadAire/Ficheros/PlanAire&CC_Eng.pdf, last access: January 21, 2022)."

19. - line 197: mention that the airport is presented on Fig2d with a triangle.

Thanks. Changed accordingly.

20. - line 203: what does mean the 1/2 subscript on the different directions? it is also present in Fig A3, but not in its caption.

The wind regimes are divided into two opposite fields and thus, the ½ subscript represents the half of the whole wind regimes, e.g., $NE_{1/2}$ for wind direction of 0º-90º and $SW_{1/2}$ for 180º-270º. The caption of Fig A3 (now is Fig7 in the updated manuscript) has been updated.

21. - line 227: suggestion "The ML-estimated emission strengths *for Madrid* are presented in Figure 4."

Thanks. Changed accordingly.

22. - line 228: suggestion "However, for weekends, the northeastern regions *(close to the airport)*, far away from the city center, are the main sources,..."

Thanks. Changed accordingly.

23. - line 259: "The NO2 emission estimate in the urban area *of Madrid* is about..."

Thanks. Changed accordingly.

24. - lines 260-267: add more discussion of other lockdown studies (see above some suggestions). (Do the same for Riyadh)

Thanks. The discussion has been added to the manuscript. Please see our reply for Q5.

25. - line 282: same as said before "This difference might be due to the different study period and methods"--> this is too general. some investigations should be performed to give at least some error estimate/ quantification of impact of some of the choices made here.

Thanks for this important comment. Another section of uncertainty analysis (including different choice of α, τ, wind field segmentation and wind on vertically and horizontal dimension) has been added to the manuscript.

26. - line 305: "But, it is can be applied to other key gases such as carbon dioxide or methane, and in other regions" --> as mentioned for the introduction, for me this is too general/optimistic. In my view carbon dioxide or methane have very different life-times and the urban emissions should be disentangled from the background. If you feel differently, please provide some supporting comments to your sentence.

Thanks for this comment. The referee is right the long-lived species, such as $CO_2$ or $CH_4$ differ from short-lived species, such as $NO_2$, since the background must be considered and removed to obtain the signal. We then modified the sentences accordingly:

"Our easy-to-apply method has successfully probed its consistency and reliability in two contrasting examples (Riyadh and Madrid). However, application in some areas with complicated emission source distribution and topography might not be feasible. The varying decay time for short-lived species in different regions and seasons is another important factor affecting the estimates of emissions. We plan to include these refinements in future studies to reduce the uncertainties of both the wind-assigned

anomaly method and the ML approach. The spatial distributions of estimates generally show checkerboard-like structures. We assume that these structures indicate that the inversion attempts to resolve fine structure which is poorly constrained by the observation. When we converge to a stable solution with minimal bias, we are confident that spatially averaged retrieved emissions are more realistic. It is our hope that the method presented here can be applied to other key gases such as carbon dioxide or methane for which the background concentration needs to be considered, and in other regions. Meanwhile, the powerful ML framework might allow to investigate related questions, perhaps a joint estimation of $NO_2$ lifetime and emission strength would be possible."

References: --------------------

Levelt, P. F., Stein Zweers, D. C., Aben, I., Bauwens, M., Borsdorff, T., De Smedt, I., Eskes, H. J., Lerot, C., Loyola, D. G., Romahn, F., Stavrakou, T., Theys, N., Van Roozendael, M., Veefkind, J. P., and Verhoelst, T.: Air quality impacts of COVID-19 lockdown measures detected from space using high spatial resolution observations of multiple trace gases from Sentinel-5P/TROPOMI, Atmos. Chem. Phys. Discuss. [preprint], https://doi.org/10.5194/acp-2021-534, in review, 2021.

Bauwens, M., Compernolle, S., Stavrakou, T., Müller, J.-F., van Gent, J., Eskes, H., et al. (2020). Impact of coronavirus outbreak on NO2 pollution assessed using TROPOMI and OMI observations. Geophysical Research Letters, 47, e2020GL087978. https://doi.org/10.1029/2020GL087978

Georgoulias, A. K., van der A, R. J., Stammes, P., Boersma, K. F., and Eskes, H. J.: Trends and trend reversal detection in 2 decades of tropospheric NO2 satellite observations, Atmos. Chem. Phys., 19, 6269–6294, https://doi.org/10.5194/acp-19-6269-2019, 2019.

---

## Author Comment (AC2)

Response to Referee #2
We would like to thank reviewer #2 for taking the time to review this manuscript and for providing valuable, constructive feedback and corresponding suggestions that helped us to further improve the manuscript.
In this authors' comment, all the points raised by the reviewer are copied here one by one and shown in blue color, along with the corresponding reply from the authors in black.

The manuscript presents an interesting new method for the determination of spatially resolved emission maps around megacities based on TROPOMI NO2 observations, wind fields from meteorological models, and machine learning techniques. The study matches the scope of AMT. Before publication, however, major additions/extensions are needed.

The paper presents results of the new approach exemplarily for Riyadh and Madrid and states that the method "works properly and is reliable" (line 283).
However, the resulting emission maps reveal strong artefacts which are not at all mentioned in the paper.
A critical evaluation/discussion of shortcomings, artefacts, problems, and uncertainties of the proposed approach is missing.
**Major concerns:**
1. The presented results reveal several artefacts:
   a. several pixels of quite high emissions over regions without obvious NOx sources, e.g. at 25.05°N, 46.45°E and 25.25°N, 46.45°E (Figs. 1d, 3a).
   There is neither a significant NOx emission over this area reported in Beirle et al., 2019, nor is there a local enhancement in the NO2 column (Fig. A4).

Figure R1 presents terrain map and spatial distribution of $NO_2$ columns on a fine resolution. Several highways cross the region at 25.05°N, 46.45°E and 25.25°N, 46.45°E (in yellow) and there are some residential areas scattered in between, especially on the west of Road 535, like Malham city. There are several ready-mix concrete companies in the blue rectangle region on the east of Road 535. Mean local enhancements can be seen from the TROPOMI observations, though they are weaker than those near to the city center. However, the normalized temporal variability in these regions is far larger than that in the city center. This is maybe what is accounting for the enhanced emissions in this region, following Figure R2.

[Figure]

Figure R1: Terrain map (left) and TROPOMI tropospheric $NO_2$ column on a regular latitude-longitude grid with 0.05° spacing (right) in Riyadh. Red rectangles represent the area between 25.05°N - 25.25°N and 46.30°E - 46.60°E.

[Figure]

Figure R2: Time series of variations (tropospheric NO₂/mean value) for the area in center (24.5°N - 24.9°N and 46.6°E - 47.0°E) and for the rectangle area in Figure R1 (25.05°N - 25.25°N and 46.30°E - 46.60°E).

> b. a large extended area of positive emissions north of Madrid (>40.7°N), where CAMS emissions are close to zero.
> While values of individual pixels still look relatively low in the color coded image, the integrated emissions >40.7°N are still considerable, and I do not think that these emissions are real.

When looking at the TROPOMI observations at a fine spatial resolution, there are considerable local enhancements north of Madrid (>40.7°N) (Figure R3). The region is crisscrossed with highways and roads, connecting small cities or villages, like Segovia, Carbonero el Mayor, Navalmanzano along the highway A-601. The variations in two regions are different (Figure R4) in different seasons, which indicates different emission sources (e.g., biomass burning, construction work, fertilization, etc.). City center region shows some bias low in May-September and bias high in December-March. Thus, some positive emission north of Madrid can be expected. But the referee is correct that there are errors existing. We applied a simple empirical model for calculating enhanced NO₂. It is determined by angle $\alpha$ which is an empirical value, and average decay time $\tau$ whose seasonal and spatial variability is not considered. These two parameters introduce unavoidable uncertainties (see Q5 below).

[Figure]

Figure R3: similar to Figure R1 but for Madrid. Orange rectangle represents the area in north of Madrid (>40.7°N)

[Figure]

Figure R4: Time series of variations (tropospheric $NO_2$/mean value) for the area in center 40.3°N - 40.5°N and 3.9°W – 3.5°W) and for the grids with dominant emissions (>1.0E23 molec./s) in north of Madrid (>40.7°N).

[Figure]

Figure R5: estimated emissions in Madrid. The grids with dots represent the emission rate > 1E23 molec./s in north of Madrid (>40.7°N).

    c.   generally "checkerboard-like" structures in the maps for data subsets.
        This indicates a problem with the method that involves solving a linear equation iteratively. It seems that initial deviations are overcompensated in the next neighbor, than undercompensated in the 2nd next neighbor, and so on, indicating an instable system with oscillating values in the solution. I think this effect is a known problem for inverse problems, and the authors might check whether they find standard procedures for avoiding or supressing these oscillations.
        In any case, the authors have to clearly describe the artefacts and discuss possible reasons.

The Gradient Descent (GD) approach tends to obtain the minimal biased (loss function) between true values (satellite observations) and modeled values (wind-assigned anomalies). The loss function is decreasing during the iteration process (see Figure R6). In our study, up to 80000 iterations are performed until reaching the minimal bias. Thus, there are no oscillating values in the solution.

The uncertainties are discussed in an additional subsection in the manuscript.

[Figure]

Figure R6: Loss and variance as a function of number of iterations.

As already demonstrated, the temporal variability contributes significantly to the emissions. What would be most interesting would be for follow-up work relating to this (Wang et al., 2021). Is it being caused by fires, but new urbanization, or something else? However, thanks to the deeper review called for by the referee, we now are more confident that these results actually are emissions, and not just noise.

The sentences have been added to the conclusion in the updated manuscript:

"We assume that the checkerboard-like structures indicate that the inversion attempts to resolve fine structure which is poorly constrained by the observation. When we converge to a stable solution with minimal bias, we are confident that spatially averaged retrieved emissions are realistic."

Wang, S., Cohen, J. B., Deng, W., Qin, K., & Guo, J. (2021). Using a new top-down constrained emissions inventory to attribute the previously unknown source of extreme aerosol loadings observed annually in the Monsoon Asia free troposphere. Earth's Future, 9, e2021EF002167. https://doi.org/10.1029/2021EF002167.

2. The authors should be more careful about the usage of the terms "NO2" and "NOx". Emissions are sometimes referred to as NO2 and sometimes as NOx in the manuscript. Clarify the issue of NOx = NO + NO2 in the beginning (emissions should refer to NOx, while TROPOMI only measures NO2). Specify how you account for the missing NO in the NOx budget. Note that there are more "oxides of nitrogen" (line 36) such as NO3, N2O5 or N2O, which are not included in NOx.

Thank the referee for pointing this mistake which might mislead readers. We have clarified the issue of NOx = NO + NO2 in the abstract and in the introduction. The misleading phrase of "oxides of nitrogen" is removed as well. We specify the emission of NO2, and the use of NOx is corrected to NO2.

3. In the introduction the authors give a quite high range for the lifetime of NOx of 1-12 hours (should be "tropospheric" rather than "atmospheric" lifetimes in line 43).

Thanks. We changed the "atmospheric" to "tropospheric".

> However, later they just use one fixed value, ignoring probable seasonal and spatial variability of the lifetime. This simplification has to be stated clearly and the impact on the resulting emission maps should at least be investigated with some simple case studies.

The referee is right that the probable seasonal and spatial variability of the lifetime exist, but it is not considered in the study. To make it clear, this information is added to Section 2.2:

> "$v$ is the wind speed from ERA5 and $\tau$ is the lifetime/decay time for $NO_2$. For simplification, seasonal and spatial variability of lifetime is not considered, and empirical values based on Beirle et al. (2019, 2011), i.e., fixed values of 4 hours for Riyadh and 7 hours for Madrid, are used in this study."

The choices of $\tau$ are adopted from the studies of Beirle et al. (2011, 2019). We discussed the uncertainty of the choices of $\tau$ as the $\alpha$ in the additional section 4 in the updated manuscript.

> "The angle ($\alpha$) of the emission cone is an empirical value, so as the lifetime/decay time ($\tau$) for $NO_2$. They can introduce uncertainties and thus, different values for $\alpha$ and $\tau$ are used to investigate their impacts on emissions. The spatial patterns of the estimates with using different $\alpha$ or $\tau$ are quite similar. The absolute values of emission rate increase with the increasing $\alpha$ (see Figure 7-left). A change of 10º in $\alpha$ introduces a difference of less than 3.2%. A decrease of 1.5% is observed when using $\alpha = 50º$, and an increase of 1.4% is observed for $\alpha = 70º$, as compared to $\alpha = 60º$. The increasing values of $\tau$ result in lower estimates (see Figure 7-right). With respect to the result obtained with $\tau = 4h$, the estimate increases by ~42% for $\tau = 3h$, and it decreases by ~20% for $\tau = 5h$.

[Figure]

Figure 7: Estimated emissions under different cone angle α (left) and $NO_2$ lifetime τ (right) based on TROPOMI data in Riyadh in 2019."

> 4. The study makes several simplifications such as constant lifetime, constant wind field, no consideration of seasonal effects (which might correlate with wind direction and thus would directly affect the wind-assigned anomaly). A discussion of the impact of these simplifications, and in general an error discussion is missing.

The referee is right that we simplified the method by using constant lifetime ($\tau$) and opening angle ($\alpha$). For wind field, the time-resolved variation has been considered based on the ERA5 wind which has a temporal resolution of 1h. Only the spatial variation of the wind field across the scene is ignored for limiting the computational effort and this impact is expected to be small (1.9% in Riyadh and -1.3% in Madrid). We have added a section (Section 4) about the uncertainty analysis as the referee suggested. The constant wind field in spatial distribution is discussed as well.

5. Finally, it is not clear to me what exactly would be the benefit of the proposed method. Quantifying megacity emissions is of course a valid goal, but this could be done with simpler methods as well. So the "extra" of the proposed method would be the generation of spatially resolved emission maps. For this purpose, a discussion of uncertainties and "reliability" of emission values for individual pixels is required. In addition, the authors should indicate concrete applications for the derived emission maps.

Thanks to the referee for this comment and raising up the question. This study proposes a methodological simple approach to obtain the emission strengths and their spatial patterns. The simplicity is one of the benefits. The method is based on simple assumptions and real observations and doesn't address the complex inverse modelling approaches. The applications are multiple, one of the most important is to determine the location and quantification of emission sources. Since it's based on actual observations, it helps to support the development and implementation of the control and mitigation policies.

We agree that several simplifications (as commented in Q4) introduce uncertainties. However, the use and benefits of this new approach are also shown. This method does quite a good job at identifying areas which have high emissions variability, and/or were missed by previous studies. This would likely be of use in regions without detailed bottom-up inventories such as in the Global South, or in areas with a rapidly changing emissions profiles, such as those undergoing significant biomass burning.

We have added another section to investigate the uncertainties. Consideration of seasonal and spatial variations of $\alpha$ and $\tau$ will be future target, which might help to reduce the biases.

**Further comments:**
6. Selection of sample locations: application of the method for Riyadh is quite straightforward due to the good observation conditions, as well as the split of wind directions almost equally in two opposite directions. But I wonder how the method should work for Madrid, as there is basically one dominating wind direction. So the wind-assigned anomaly can definitely tell you where Madrid is located, but with this "unimodal" wind pattern, I really wonder what additional information on spatial distribution of sources should be gained. There might be other megacities where the approach might be more promising.

The uncertainty of the wind segmentation has been discussed in Section 4.2. Approximate 12.5% and 8.6% changes are observed if we use a different wind segmentation of NE/SW for Riyadh and SE/NW for Madrid, respectively. The spatial distribution of estimates in Riyadh changes slightly when wind segmentation changes. However, some positive emissions are obtained in southwest of Madrid. Using different wind segmentation leads to different spatial distributions of estimates, especially in Madrid where the topography (e.g., land cover, altitude) is more complicated than in Riyadh.

We selected Madrid to confront our proposed method with a more problematic scene (due to the existence of a dominating wind direction, as the referee correctly points out). There are already several studies on Madrid (Bauwens et al., 2022; Levelt et al., 2022), which can be used as "reference" for evaluating the success of our method.

[Figure]

Figure 8: (a) similar to Figure 1 (d), but using SW-NE wind field segmentation for Riyadh; (b): similar to Figure 2 (d), but using SE-NW wind field segmentation for Madrid.

7. Line 70: "used to train": training is a crucial element of any ML, and I wondered here against which "truth" the ML is trained. It needs Eq. 1 to understand how the "modeled truth" is constructed that is used for training. I think that this is also an important component of this approach, that a simple downwind plume model is used to construct the NO2 distribution for the emissions from each grid pixel.

In this method, the true values (also called labels) are the TROPOMI observations. The referee is right that the "modeled truth" is used for training and obtaining the $w$, i.e., the emissions in the end.

We cannot train the true values and the sentence is wrongly stated. It has been changed to:

"In this study, the Gradient Descent (GD) approach in ML incorporating the wind-assigned method (Tu et al., 2022a, 2022b) is used to train the "modeled truth" constructed from a simple downwind plume model for the emissions on each grid pixel using space borne $NO_2$ observations, to estimate the $NO_2$ emission strengths of two (mega)cities: Riyadh (Saudi Arabia) and Madrid (Spain)".

8. Section 2.2: The authors describe eq. 1 as the "averaged distribution ... over a long time period" (Line 90), but later apply this to "daily plumes". Please clarify.

Eq. 1 might introduce biases for a single day as the choice of opening angle is either too small or too large, or lifetime is either too short or too long, or the distribution of emitted $NO_2$ is not even. However, this equation becomes reasonable for a long-term period as the biases are largely compensated.

9. Eq. 1: what should be the meaning of the division by an angle in degree? I think this cannot be correct - is there a sin or cos missing? Otherwise, please clarify the units of all components of Eq. 1.

The opening angle of the plume has a unit of rad. We have updated this in the text.

The units of all components are clarified according to referee's comment.

10. Line 115: I understand the motivation for choosing log(wk) as proxy for wk. However, this drastically modifies the weight of the different pixels with focus on very low emission values. The satellite measurements, on the other hand, have highest signal to noise for the pixels with very high emission values.

This has to be discussed. Have you tried to run the algorithm directly with wk instead?
This will of course result in some negative emission values, but the results for strong sources like powerplants might be more reliable.
Please also specify the exact procedure: are Eqs. 5-7 applied for wk or actually for log(wk)? If the latter is the case, then there should also be log(wk) written in all equations, plus an additional equation indicating how final emissions are derived (perhaps trivial, but I think very helpful for understanding what was done exactly).

Thanks for this important comment. We have tried to run the algorithm directly with wk, i.e., linear emission (see Figure R7). The spatial patterns of estimates (except the negative ones) are comparable to our previous results. The total emission rates decreased by 30% for Riyadh and 2% for Madrid, respectively. More negative emissions are modelled for Riyadh in south region than for Madrid. This means, using log(wk) does not drastically modify the weight on the pixels with low emission values.

We believe the log(wk)-based reconstruction is superior despite of assigning higher weights to small positive emissions, because it excludes unphysical negative emissions, which occur in the linear reconstruction.

log(wk) is just a proxy in the ML process, but the real wk, i.e., exp(log(wk)) is used in Eqs. 5-7.

[Figure]

Figure R7: Estimated emission in Riyadh and Madrid without using log for wk (a-b) and using log (c-d) (results based on data in 2019).

[Figure]

Figure R8: Ratio of estimated rates between using log and linear for Riyadh and Madrid.

**11. Line 115: Where is the initial epsilon coming from?**

The initial epsilon is an empirically random value, or in other words it can be any number. We use 1E26 here. This value will be later scaled by the weights w.

**12. Line 132: If the outer ring should be skipped in order to skip edge effects, the initial study area must be (n+2)x(m+2), since one pixel on each side has to be skipped.**

Thanks! We have corrected this mistake.

**13. Line 152: In addition, for Riyadh the separation into two wind regimes suggests itself.**

Thanks to the referee. The sentence has been added in the manuscript:

"The typical two wind regimes presented in Riyadh favors the applicability of the wind-anomaly method and is another reason of choosing it for the work."

**14. Line 279: I would not agree that the spatial patterns agree very well. There are some artefacts present in the presented emission maps, and the conclusions should reflect this.**

This sentence has been changed to "The spatial pattern of the estimated emission strengths on the main sources near the city center agrees with the results from Beirle et al. (2019) as well".

**15. I'm no native English speaker myself. However, several sentences and formulations sound strange to me. I would recommend careful language check after dealing with the requested modifications/extensions.**

Thanks for pointing this out. We have done our best again to improve the language.

**Technical issues:**
**16. Line 49: etal. -> et al.**

Thanks. Corrected.

**17. Line 50: TROPOMI acronym explained twice.**

Thanks. Corrected.

**18. Line 80: To which area are the 910,000 measurements refer to? I think it would be more useful to give a typical number how many measurements there are per 0.1° grid pixel.**

The information of the study areas has been added to the text. Additionally, the total amounts of measurements have been updated as the study period is extended. The sentence below has been added to section 2.1:

"There are nearly 1,380,000 in Riyadh (23.5ºN – 25.5ºN; 46ºE – 47.5ºE) and 930,000 measurements in Madrid (39.5ºN – 41.5ºN; 4.5ºW – 3ºW) of good quality over three years."

The following figures represent the amounts of TROPOMI NO$_2$ measurements in each 0.1º grid pixel and they are added to the appendix. The following sentence has been added to section 2.1 as well:

"The amounts of TROPOMI measurements in each 0.1º grid pixel is distributed evenly with a number range of 4500-5000 in Riyadh, whereas larger difference is observed in Madrid with a number range of 2300-4000 (see Figure A- 1)."

[Figure]

**Figure A- 1: Amount of TROPOMI measurements in each 0.1° grid pixel for Riyadh and Madrid during May 2018 – August 2022.**

19. all emission maps: The emissions are given in molecules per second, but implicitly refer to the chosen grid. I.e., for 0.05° grid, numbers would be only 1/4 of the presented values. Emission maps should thus be provided as densities (emissions per time per area).

Thanks for this comment. The referee is right that emission densities would be grid-independent and perhaps a superior choice, but we believe we have handled the required scaling of emissions per grid box when transitioning between the two different grid sizes correctly.

20. all maps: please choose the same lat/lon range for all plots for Riyadh and Madrid, respectively.

The terrain maps have been updated by adding latitude and longitude to the figures (as suggested by the other referee). Larger areas are preferred as it can help readers to better understand the topography of the whole area. The study areas are illustrated by using white rectangles. It is difficult to specify the boundary of the study area in the zoom version for Madrid in Figure A- 11. We would like to keep this figure for showing the land use of the Madrid as an additional information.

[Figure]

21. Caption Fig. 1: 0.1° is coarser than most TROPOMI pixels, thus the data is not "oversampled".

The "oversampled" seems bringing misleading information. The sentence has been changed to "Data in (a), (b), and (d) are gridded on a regular latitude-longitude grid with 0.1° spacing".

22. x labels of Fig. 1 (c), 2 (c), 3 (f): "TROPOMI tropospheric NO2" is misleading here, as the shown quantity is a difference (or anomaly).

The referee is right that the x labels in these figures represent the wind-assigned anomalies derived from TROPOMI tropospheric $NO_2$. This information can be obtained from the subtitle of the figures. The x and y labels represent the data sets from where the wind-assigned anomalies are derived. We have added this information to the figures' caption.

23. Figs. 3 (c) and 4 (c): There are several pixels where weekend emissions are higher than on weekdays. Thus the colorbar in (c) should be symmetric around 0, including negative values as well.

Thanks. The figures have been updated.